



# Top-down and Bottom-up aerosol-cloud-closure: towards understanding sources of uncertainty in deriving cloud radiative flux

Kevin J. Sanchez[1,2], Greg C. Roberts[1,2], Radiance Calmer[2], Keri Nicoll[3,4], Eyal Hashimshoni[5], Daniel Rosenfeld[5], Jurgita Ovadnevaite[6], Jana Preissler[6], Darius Ceburnis[6], Colin O'Dowd[6], Lynn M. Russell[1]

[1]Scripps Institution of Oceanography, University of California, San Diego, CA;
[2]Centre National de Recherches Météorologiques, Toulouse, France;
[3]Department of Meteorology, University of Reading, UK;
[4] Department of Electronic and Electrical Engineering, University of Bath, UK
[5]The Hebrew University of Jerusalem, Israel;
[6]School of Physics and Centre for Climate and Air Pollution Studies, National University of Ireland Galway, Ireland;

*Correspondence to*: Kevin J. Sanchez (kjsanche@ucsd.edu)

**Abstract.** Top-down and bottom-up aerosol-cloud-radiative flux closures were conducted at the Mace Head atmospheric research station in Galway, Ireland in August 2015. This study is part of the BACCHUS (Impact of Biogenic versus Anthropogenic emissions on Clouds and Climate: towards a Holistic UnderStanding) European collaborative project, with the goal of understanding key processes affecting aerosol-cloud-radiative flux closures to improve future climate predictions and develop sustainable policies for Europe. Instrument platforms include ground-based, unmanned aerial vehicles (UAV)[1], and satellite measurements of aerosols, clouds and meteorological variables. The ground-based and airborne measurements of aerosol size distributions and cloud condensation nuclei (CCN) concentration were used to initiate a 1D microphysical aerosol-cloud parcel model (ACPM). UAVs were equipped for a specific science mission, with an optical particle counter for aerosol distribution profiles, a cloud sensor to measure cloud extinction, or a 5-hole probe for 3D wind vectors. UAV cloud measurements are rare and have only become possible in recent years through the miniaturization of instrumentation. These are the first UAV measurements at Mace Head. ACPM simulations are compared to in-situ cloud extinction measurements from UAVs to quantify closure in terms of cloud radiative flux. Two out of seven cases exhibit sub-adiabatic vertical temperature profiles within the cloud, which suggests that entrainment processes affect cloud microphysical properties and lead to an overestimate of simulated cloud radiative flux. Including an entrainment parameterization and explicitly calculating the entrainment fraction in the ACPM simulations both improved cloud-top radiative closure. Entrainment reduced the difference between simulated and observation-derived cloud-top radiative flux ($\delta RF$) by between 30 W m$^{-2}$ and 40 W m$^{-2}$. After accounting for entrainment, satellite-derived cloud droplet number concentrations (CDNC) were within 30% of simulated CDNC. In cases with a well-mixed boundary layer, $\delta RF$ is less than 25 W m$^{-2}$ after accounting for cloud-top entrainment, compared to less than 50 W m$^{-2}$ when entrainment is not taken into account. In cases with a decoupled boundary layer, cloud microphysical properties are inconsistent with ground-based aerosol measurements, as expected, and $\delta RF$ is as high as 88 W m$^{-2}$, even after accounting for cloud-top entrainment. This

---

[1] The regulatory term for UAV is Remotely Piloted Aircraft (RPA).



work demonstrates the need to take in-situ measurements of aerosol properties for cases where the boundary layer is decoupled as well as consider cloud-top entrainment to accurately model stratocumulus cloud radiative flux.

## 1 Introduction

One of the greatest challenges in studying cloud effects on climate are that the clouds are literally out of reach. Many ground-based measurement sites have a long historical record that are useful for identifying climatological trends, however, it is difficult to quantify such trends in cloud microphysical and radiative properties at these stations based solely on remote sensing techniques such as radar and lidar. *In-situ* aerosol measurements at the surface are often used to estimate cloud properties aloft, but the simulations used to estimate above surface conditions require many idealized assumptions such as a well-mixed boundary layer and adiabatic parcel lifting. Satellites have the advantage to infer cloud properties over a much larger area than ground-based observations; however, they can only see the upper most cloud layer and satellites need *in-situ* observations to improve their retrievals. In this study, we combine ground-based and airborne measurements with satellite observations to determine cloud radiative properties and compare these results to an aerosol-cloud parcel model (ACPM) to identify sources of uncertainty in aerosol-cloud interactions.

The atmospheric research station at Mace Head has been a research platform for studying trace gases, aerosols and meteorological variables since 1958 [*O'Connor et al.*, 2008]. The station is uniquely exposed to a variety of air masses, such as clean marine air and polluted European air. Over the long history of observations and numerous field-campaigns held at the Mace Head research station, few airborne field experiments have been conducted. During the PARFORCE campaign in September 1998, aerosol and trace gas measurements were made to map coastal aerosol formation [*C. D. O'Dowd et al.*, 2001]. During the second PARFORCE campaign in June 1999, measurements of sea spray plumes were made on an aircraft installed with a Lidar [*Kunz et al.*, 2002]. In the NAMBLEX campaign in August 2002, flights were conducted to measure aerosol chemical and physical properties in the vicinity of Mace Head [*Coe et al.*, 2006; *Heard et al.*, 2006; *Norton et al.*, 2006]. None of the research flights thus far have studied aerosol-cloud interactions and cloud radiative properties at Mace Head.

For ground-based observations, it is often assumed that measured species are well-mixed throughout the boundary layer. Often this assumption is valid and many observational studies have shown that models which use ground-based measurements can accurately simulated cloud droplet number concentrations (CDNC) [*Conant et al.*, 2004; *Fountoukis et al.*, 2007; *Russell and Seinfeld*, 1998], making bottom-up closure a viable method for predicting cloud properties. Closure is defined here as the agreement between observations and model simulations of CDNC and cloud-top radiative flux. This well-mixed boundary layer simplification, however, has been shown to be inaccurate in many field experiments (e.g., the Atlantic Stratocumulus Transition Experiment (ASTEX) [*Albrecht et al.*, 1995]; and the Aerosol Characterization Experiments, ACE1 [*Bates et al.*, 1998] and ACE2 [*Raes et al.*, 2000]. Previous studies at Mace Head have shown that decoupled boundary layers were observed with scanning backscatter lidar measurements [*Kunz et al.*, 2002; *Milroy et al.*, 2012]. Such decoupled layers often contain two distinct cloud layers, distinguished as a lower layer within the well-mixed surface layer and a higher decoupled residual layer between the free troposphere





and surface layer [*Kunz et al.*, 2002; *Milroy et al.*, 2012; *Stull*, 1988]. General characteristics associated with decoupled boundary layers are a weak inversion, a decrease in aerosol concentration relative to the surface layer, and more commonly occurring in relatively deep marine boundary layers ( > 1400 m) [*Jones et al.*, 2011]. Dall'Osto et al [2010]

showed the average height of the surface mixed layer, over Mace Head, varies from 500 m to 2000 m, and the decoupled layers have heights ranging from 1500 m to 2500 m. Marine boundary layer decoupling is often seen in the tropics and has been attributed to processes that involve cloud heating and surface cooling as cloud warming can result from cloud-top entrainment, leading to decoupling of the boundary layer [*Albrecht et al.*, 1995; *Bates et al.*, 1998; *Bretherton et al.*, 1997]. In addition, Bretherton and Wyant [1997] have suggested that the decoupling structure is

mainly driven by an increasing ratio of the surface latent heat flux, (i.e., evaporative cooling at the surface) to the net radiative cooling within the cloud, while other factors, such as drizzle, the vertical distribution of radiative cooling in the cloud, and sensible heat fluxes, play less important roles. Turton and Nicholls [1987] used a two-layer model to show that decoupling can also result from solar heating of the cloud layer. Nicholls and Leighton [1986] suggested decoupling results from cloud-top radiative cooling and the resulting eddies do not mix down to the surface. Zhou et

al. [2015] showed that the entrainment of the dry warm air above the inversion could also be the cause. Marine boundary layer decoupling is often seen in the tropics and has been attributed to easterlies bringing air over increasing SST, which increases latent cooling and adds negative buoyancy below the cloud layer [*Albrecht et al.*, 1995]. Russell et al. [1998] and Sollazzo et al. [2000] showed that, in a decoupled atmosphere the two distinct layers have similar characteristics (e.g., aerosol and trace gases composition), but different aerosol concentrations and gradually mix with

each other, entraining air from the surface layer into the decoupled residual layer and vice versa. These previous studies show that aerosol concentrations in the upper residual layer are lower than those in the well-mixed surface layer implying an overestimation in cloud radiative flux when using ground-based aerosol measurements.

Satellite measurements of microphysical properties, such as CDNC, have the potential to be independent of ground-

based measurements, and therefore be reliable for studying decoupled clouds. Satellite estimates of CDNC have only become possible recently due to the increased resolution in measurements [*Painemal and Zuidema*, 2011; *Rosenfeld et al.*, 2014; *Rosenfeld et al.*, 2012; *Rosenfeld et al.*, 2016]. Therefore, current measurements still require ground-based validation until the method is further developed.

The focus of this manuscript is on the top-down closure between satellite retrievals and airborne measurements of cloud microphysical properties, as well as, traditional bottom-up closure coupling below and in-cloud measurements of cloud condensation nuclei (CCN), updraft, and cloud microphysical properties. *In-situ* measurements of CDNC are not available so bottom-up closure is expressed in terms of cloud-top radiative flux rather than CDNC and top-down closure of satellite CDNC is compared to ACPM simulated CDNC. The methods section describes how observations

were collected, as well as the methods for estimating CDNC with satellite measurements and calculating radiative flux with the ACPM. The results section summarizes the bottom-up and top-down closure for coupled and decoupled clouds and quantifies the differences in cloud radiative flux for cases that were affected by cloud-top entrainment.


## 2 Methods

The August 2015 campaign at the Mace Head research station (Galway, Ireland; 53.33ºN, 9.90ºW) focused on aerosol-
cloud interactions at the north eastern Atlantic Ocean by coupling ground-based *in-situ* and remote sensing
observations with airborne and satellite observations. This section summarizes the measurements used for this study
and the model used to simulate the observations.

### 2.1 Ground-based measurements

At the Mace Head research site, aerosol instruments are located in the laboratory at about 100 m from the coastline.
They are connected to the laminar flow community air sampling system, which is constructed from a 100 mm diameter
stainless-steel pipe with the main inlet at 10 m above ground level, so that samples are not impacted by immediate
coastal aerosol production mechanisms, such as wave breaking and biological activity [*Coe et al.*, 2006; *Norton et al.*,
2006; *C O'Dowd et al.*, 2014; *C. D. O'Dowd et al.*, 2004; *Rinaldi et al.*, 2009]. The performance of this inlet is
described in Kleefeld et al.[2002]. Back trajectories during the period of the experiment show that the origin of air
masses is predominantly from the North Atlantic; therefore, the air masses sampled at Mace Head generally represent
clean open ocean marine aerosol. Mace Head contains a variety of aerosol sampling instrumentation, spanning particle
diameter range of 0.02 µm and 20 µm. Size spectral measurements are performed at a relative humidity < 40% using
Naphion driers. Supermicron particle size distributions were measured using an Aerodynamic Particle Sizer (APS,
TSI model 3321, $0.5 < Dp < 20$ µm). The remaining submicron aerosol size range was retrieved from a scanning
mobility particle sizer (SMPS, $0.02 < Dp < 0.5$ µm), comprised of a differential mobility analyzer (DMA, TSI model
3071), a condensation particle counter (TSI model 3010, $Dp > 10$ nm), and a Kr-85 aerosol neutralizer (TSI 3077).
Cloud condensation nuclei (CCN) measurements were performed with a miniature Continuous Flow Stream-wise
Thermal Gradient Chamber, which measures the concentration of activated CCN over a range of supersaturations
[*Roberts and Nenes*, 2005]. During this study, the supersaturation range spanned 0.2% to 0.82%. Aerosol
hygroscopicity was calculated using κ-Köhler theory [*Petters and Kreidenweis*, 2007] with the sampled CCN
concentrations at a particular supersaturation and corresponding integrated aerosol number concentration at a critical
diameter [*Roberts et al.*, 2001]. Figure 1 shows time series of CCN spectra and aerosol number size distributions
throughout the campaign. The ground-based remote sensing measurements utilized in this study are the MIRA36, 35.5
GHz Ka-band Doppler cloud radar [*Goersdorf et al.*, 2015; *Melchionna et al.*, 2008] to obtain vertical velocity
distributions at cloud-base and the Jenoptik CHM15K ceilometer [*Heese et al.*, 2010; *Martucci et al.*, 2010] to obtain
cloud base height.

### 2.2 UAV vertical profiles

The UAV operations were conducted directly on the coast about 200 meters from the Mace Head research station.
UAVs were used to collect vertical profiles of standard meteorological variables, temperature (IST, Model
P1K0.161.6W.Y.010), pressure (Bs rep Gmbh, Model 15PSI-A-HGRADE-SMINI), and relative humidity (IST, P14
Rapid-W), as well as aerosol size distributions with an optical particle counter (OPC, Met One Model 212-2), cloud
droplet extinction [*Harrison and Nicoll*, 2014], updraft velocity at cloud base with a 5-hole probe. A list of the various





UAV flights and their instrumentation is given in Table 1. Measurement errors for the relative humidity and temperature sensors are ± 5% and ± 0.5 ℃ respectively. As RH sensors are not accurate at high RH ( > 90%), the

measured values have been scaled such that RH measurements are 100% in a cloud. At altitudes where the UAV is known to be in-cloud (based on *in-situ* cloud extinction measurements) the air mass is considered saturated (RH ~ 100%). The temperature and relative humidity sensors are protected from solar radiative heating by a thin-walled aluminum shroud positioned outside of the surface layer of the UAV. A helical cone, mounted in front of the sensors, ejects droplets to protect the sensors. The temperature measurements for both cases in which cloud-top entrainment

is explored (see section 3.2) are verified to remain in stratocumulus clouds throughout the ascents and descents, and are not affected by evaporative cooling. The temperature and relative humidity measurements were used to initialize the ACPM below cloud. The UAVs were flown individually in separate missions up to 1.5 hours and each UAV was instrumented to perform a specific science mission (referred to here as aerosol, cloud, 3D winds).

The OPC measured aerosol number size distributions in eight size bins between 0.3 and 10 µm diameter. Aerosols were sampled via a quasi-isokinetic shrouded inlet mounted on the nose of the UAV. Aerosols samples were heated upon entering the UAV ($\Delta T > 5$ K due to internal heating by the electronics), reducing the relative humidity of the sampled air to less than 60% and decreased with height ( < 50% above 150 m) before aerosol size was measured. Figure 2 shows a two-instrument redundancy cross check between ground-based APS and UAV OPC measurements

(collected between 40 m agl and 80 m agl) of aerosol sizes are in agreement ($r^2 = 0.48$).

In-cloud extinction was measured in-situ using a disposable optical cloud droplet sensor developed at the University of Reading [*Harrison and Nicoll*, 2014]. The sensor operates by a backscatter principle using modulated LED light which is backscattered into a central photodiode. Comparison of the sensor with a DMT Cloud Droplet Probe

demonstrate good agreement for cloud droplet diameters >5µm [*Nicoll et al.*, 2016].

Finally, a 5-hole probe for measuring 3-dimentional wind vectors was mounted on a third UAV. The 3D wind vectors are determined by subtracting the UAV motion given by an inertial measurement unit (IMU) from the total measured flow obtained by differential pressures in the 5-hole probe [*Calmer et al.*, 2017; *Lenschow and Spyers-Duran*, 1989;

*Wildmann et al.*, 2014]. UAV 5-hole probe measurements were collected along 6 km long straight and level legs at cloud base. Normalized cloud radar vertical velocity distributions are compared to vertical wind distributions obtained from the UAV in Figure 3. The positive updraft velocities in Figure 3 are used to initialize the ACPM to produce simulated cloud droplet size distributions throughout the depth of the cloud. The droplet distributions for each updraft velocity are averaged and weighted by the probability distribution of the measured positive velocities. Differences in

results when using the cloud radar updrafts versus the UAV 5-hole probe updrafts (Figure 3) are discussed in section 3.1.2.

### 2.3 Satellite measurements

Research flights with the UAV were conducted in conjunction with satellite overpasses to compare retrieved CDNC and maximum supersaturation ($S_{max}$) with ACPM simulated values using the Suomi NASA Polar-orbiting Partnership





satellite. The satellite estimations of CDNC and $S_{max}$ are based on methods described by Rosenfeld et al.[2014; 2012; 2016], which are briefly summarized in the following paragraph. The case selection criteria for satellite observations required the overpass to occur at a zenith angle between 0º and 45º to the east of the ground track, to have convective development that spans at least 6 K of cloud temperature from base to top (~1 km thick), and to not precipitate significantly. In-situ observations were often of thin clouds (< 1 km thick), and the satellite observations consist
primarily of the more developed clouds in the same system.

  To obtain CDNC, cloud droplet effective radius profiles were extracted from the Suomi NASA Polar-orbiting Partnership satellite. Figure 4 shows an image from the Suomi visible infrared imaging radiometer suite on 21 August overlapped on a map of western Ireland. The vertical profile in figure 4 shows satellite retrieved and ACPM simulated
effective radius.  To estimate the CDNC, the satellite effective radius (Figure 4) is first converted to mean volume radius ($r_v$) using a linear relationship [*Freud et al.*, 2011]. Next, it is assumed that any mixing that occurred between the cloud and cloud-free air was inhomogeneous; this implies that the actual $r_v$ is equal to the adiabatic $r_v$. CDNC can be calculated by dividing the adiabatic water content in the cloud by $r_v$ [*Beals et al.*, 2015; *Rosenfeld et al.*, 2012]. The cloud base height and pressure was used to calculate the adiabatic water content. Cloud base height and pressure
were obtained from the height of the NCEP reanalysis of the cloud base temperature, as retrieved from satellite. The cloud base height was validated against the ceilometer. Freud et al. [2011] showed that the inhomogeneous assumption resulted in an average over-estimate in CDNC of 30%, so the CDNC is reduced by 30% to account for the bias with the assumption. Finally, to calculate $S_{max}$ the cloud base updraft velocity, from the UAV or cloud radar, is needed and when paired with the CDNC, it can be used to empirically calculate $S_{max}$ [*Pinsky et al.*, 2012; *Rosenfeld et al.*, 2012].
The methodology was validated by Rosenfeld et al. [2016].

### 2.4 Aerosol-cloud parcel model simulations

  A detailed description of the aerosol-cloud parcel model (ACPM) is presented in Russell and Seinfeld [1998] and Russell et al. [1999]. The ACPM is based on a fixed-sectional approach to represent the (dry) particle size domain, with internally mixed chemical components and externally mixed types of particles. Aerosols are generally internally
mixed at Mace Head owing to lack of aerosol sources. The model employs a dual moment (number and mass) algorithm to calculate particle growth from one size section to the next for non-evaporating compounds (namely, all components other than water) using an accommodation coefficient of 1.0 [*Raatikainen et al.*, 2013]. The dual moment method is based on Tzivion et al. [1987] to allow accurate accounting of both aerosol number and mass, and incorporates independent calculations of the change in particle number and mass for all processes other than growth.
The model includes a dynamic scheme for activation of particles to cloud droplets. Liquid water is treated in a moving section representation, similar to the approach of Jacobson et al. [1994], to account for evaporation and condensation of water in conditions of varying humidity. In sub-saturated conditions, aerosol particles below the cloud base are considered to be in local equilibrium with water vapor pressure (i.e., relatively humidity < 100%).

Coagulation, scavenging, and deposition of the aerosol were included in the model but their effects are negligible given the relatively short simulations used here (<2 h) and low marine total aerosol particle concentrations (<500 cm³;





Dp > 10 nm). Aerosol hygroscopicity as a function of size (and supersaturation) is determined from CCN spectra and aerosol size distributions as mentioned in Section 3.1, and is used as model input. The ACPM is also constrained by measured temperature profiles, cloud base height, and updraft velocity distribution (Figure 3). The in-cloud lapse rate

is assumed to be adiabatic, unless specified otherwise, so simulation results represent an upper bound on CDNC and liquid water content that is unaffected by entrainment. To account for release of latent heat in the cloud, the vertical temperature gradient is calculated as $dT = (gwdt + Ldq_l)/c_p$, where $dT$ is change in temperature for the vertical displacement of an air parcel, $g$ is acceleration due to gravity, $w$ is updraft velocity at cloud base, $dt$ is time step, $L$ is latent heat of water condensation, $q_l$ is liquid water mixing ratio, and $c_p$ is specific heat of water [Bahadur et al., 2012].

A weighted ensemble of positive updraft velocities measured with the cloud radar and UAV 5-hole probe were applied to the ACPM [Sanchez et al. 2016].

The simulated cloud droplet size distribution is used to calculate the shortwave cloud extinction. Cloud extinction is proportional to the total droplet surface area [Hansen and Travis, 1974; Stephens, 1978] and is calculated from,

$$\sigma_{ext} = \int_0^\infty Q_{ext}(r)\pi r^2 n(r)\, dr \tag{1}$$

where r is the radius of the cloud droplet, $n(r)$ is the number of cloud droplets with a radius of r, and $Q_{ext}(r)$ is the Mie efficiency factor, which asymptotically approaches 2 for water droplets at large sizes (r > 2 um).

Finally, the radiative flux (RF) is calculated as RF = $\alpha Q$, where $Q$ is the daily-average insolation at Mace Head and $\alpha$

is the cloud albedo. $\alpha$ is estimated using the following equation [Bohren and Battan, 1980; Geresdi et al., 2006]

$$\alpha = \frac{(\sqrt{3}(1-g)\tau)}{(2+\sqrt{3}(1-g)\tau)}, \tag{2}$$

where $\tau$ is the cloud optical depth defined as

$$\tau = \int_0^H \sigma_{ext}(h)\, dh; \tag{3}$$

and $H$ is the cloud height or thickness and $g$, the asymmetric scattering parameter, is approximated as 0.85 based on

Mie scattering calculations for supermicron cloud drops. RF is calculated for both, simulated cloud extinction and measured UAV extinction.

### 3 Results/Discussion

### 3.1 Closure of CDNC and cloud-top radiative flux

For this study, closure is defined as the agreement between observations and model simulations of CDNC and cloud-

top radiative flux. In-situ measurements of clouds were made by UAVs on 13 days during the campaign. Of these, a subset of six are chosen here for further analysis, which includes comparison with satellite CDNC as well as simulation of cloud properties with the ACPM (Table 2). The remaining days with UAV measurements did not contain sufficient cloud measurements for analysis. A satellite overpass occurred on each of the six days, however only 4 of the days contained clouds that were thick enough to analyze with the satellite. The 10 August cases experienced a light drizzle,

so ACPM simulations were not conducted for this case, however analysis with satellite imagery was still conducted. On 5 August, two cloud layers were examined, for a total of 7 case studies shown in Table 2. Aerosols were



occasionally influenced by anthropogenic sources, however, the cases shown consist of aerosol of marine origin with concentrations under 1000 cm$^{-3}$ (Figure 1).

### 3.1.1 Ground-based measurement closure

The columns in Table 2 represent the different cases for both clouds that were (a) coupled with and (b) decoupled from the surface BL ("C" and "D" in case acronym, respectively). The first row in Table 2 includes the state of atmospheric mixing, the date, the type of cloud present, and the acronym used for each case. The top portion of Table 2 consists of *in-situ* airborne measurements, the bottom portion presents ACPM simulation results and their relation to *in-situ* cloud extinction and satellite-retrieved observations. The ground-based *in-situ* measurements in Table 2

include the Hoppel minimum diameter[2] ($D_{min}$), as well as the aerosol concentration of aerosol with diameters greater than the Hoppel $D_{min}$ and the inferred in-cloud critical supersaturation ($S_c$) [Hoppel 1979]. The aerosol particles with diameters greater than the Hoppel $D_{min}$ have undergone cloud processing and are used here to estimate the CDNC. For each of the case study days, Figure 5 demonstrates the aerosol size distribution measurements, from the SMPS and APS, that are used to find the Hoppel $D_{min}$, Hoppel CDNC and used to initialize the ACPM. Figure 6 shows Hoppel-

based CDNC estimates are within 30% of simulated CDNC for the 7 cases. The presence of the Hoppel minimum occurs on average at 80 nm diameter throughout the campaign (Figure 1b, 5) implying in-cloud supersaturations near 0.25 % using a campaign averaged hygroscopicity (K) of 0.42, which is in agreement with K values observed in the North Atlantic marine planetary boundary layer in Pringle et al. [2010].

### 3.1.2 UAV measurements closure

Figure 7 displays vertical profiles of meteorological parameters, as well as OPC aerosol number concentration ($N_{OPC}$; $Dp > 0.5$ µm) and cloud extinction from two flights (23 and 27) on 11 August. The UAV used on flight 23 (conducted between 12:00 UTC and 12:47 UTC), contained the cloud sensor for cloud extinction measurements and flight 27 (conducted between 16:58 UTC and 17:33 UTC) contained the OPC for droplet size distribution measurements. During this time period the cloud base reduced from 1200 m on flight 23 to 980 m on flight 27, but cloud depth

remained approximately the same. In the OPC vertical profiles, in Figure 7d, an aerosol layer is shown above the cloud at ~1400 m. OPC measurements are removed inside cloud layers (as aerosol data is contaminated by cloud droplets), hence the gap in OPC data in Figure 7d. The OPC and temperature measurements, in Figure 7a and d, are used to show if the boundary layer was coupled (well-mixed) or if it was decoupled. The state of the boundary layer and the OPC and temperature measurements are further discussed at the end of this section. The observed temperature and

relative humidity profiles, in Figure 7a and b, are also used to initialize the ACPM. In-situ cloud extinction measurements, in Figure 7c, are then compared to the ACPM simulated cloud extinction (Figure 8c).

Figure 8a, c and e present the observed and simulated adiabatic cloud extinction profile for three of the case studies (C11Sc, D05Sc and C21Cu)[3]. The measurements are binned into in-cloud, cloud-free, and cloud-transition (or cloud-edge) samples. Many clouds had a small horizontal extent making it difficult for the UAVs to remain in cloud as they

---

[2] The Hoppel minimum diameter is the diameter with the lowest aerosol concentration between Aitken mode and accumulation mode.

[3] C/D – coupled / decoupled; xx – date in August 2015; Sc / Cu – stratocumulus / cumulus cloud





ascended and descended in a spiral pattern. Also, high horizontal winds ($10 – 15$ m s$^{-1}$) will generally move the cloud outside the field of measurement of the aircraft very quickly. For cases where the UAV did not remain in-cloud throughout the ascent or descent, the in-cloud samples are identified as the largest extinction values at each height and are seen in the measurements as a cluster of points (Figure 8e). Since lateral mixing with cloud-free air exerts an influence near the cloud edges, the cloud-transition air is not representative of the cloud core and adiabatic simulations.

The amount of sampling within individual clouds varied from case to case, but the UAVs were generally able to make multiple measurements of the same cloud during each vertical profile. C11Sc was unique in that it involved stratocumulus clouds with a large horizontal extent, allowing the UAV to remain entirely in-cloud during the upward and downward vertical profiles around a fixed waypoint. Figure 8f shows how the difference between simulated and observed extinction ($\delta\sigma_{ext}$) is calculated throughout the cloud based on a discrete sampling of in-cloud measurements.

It is not certain that the UAV measured the cloud core for cumulus cases so $\delta\sigma_{ext}$ is an upper limit (Table 2).

All ACPM simulation results, including those in Table 2, use the cloud radar updraft velocity as input and not the 5-hole probe updraft velocity because 5-hole probe updraft velocities are not available for all cases. Nonetheless, the differences in ACPM simulated radiative flux between using the 5-hole probe and cloud radar updraft velocities

(Figure 3) is less than 3 W m$^{-2}$ for the four cases that had both measurements.

The integrated effect of $\delta\sigma_{ext}$ leads to a difference in cloud observed and simulated radiative flux ($\delta RF$) for both clouds that were coupled with and decoupled from the surface boundary layer (Table 2). Figure 9, presents a vertical profile of $N_{OPC}$ and equivalent potential temperature. OPC measurements within a thin cloud layer at ~2000 m are removed.

$N_{OPC}$ and equivalent potential temperature ($\theta_e$) clearly illustrate this decoupling as shown in an example vertical profile (Figure 9) at 900 and 2200 m.asl, with the latter representing the inversion between the boundary layer top and free troposphere. $N_{OPC}$ decreases from an average of 31 cm$^{-3}$ to 19 cm$^{-3}$ at the same altitude as the weak inversion (700-1000 m). In this study, decoupled boundary layers are often observed and aerosol number concentrations (Dp > 0.3 µm) in the decoupled layer were 44% ±14% of those measured at the ground. While $N_{OPC}$ are not directly

representative of CCN concentrations, a reduction in aerosol number with height (and potential differences in hygroscopicity) will nonetheless affect aerosol-cloud closures, and ultimately, the cloud radiative properties. Similarly, Norton et al. [2006] showed results from the European Centre for Medium-Range Weather Forecasts (ECMWF) model re-analysis in which surface winds at Mace Head are often decoupled from synoptic flow and, therefore, the air masses in each layer have different origins and most likely different aerosol properties.

Consequently, the CCN number concentrations measured at the surface do not represent those in the higher decoupled cloud layer, which ultimately dictates cloud radiative flux in the region and $\delta RF$ in Table 2. While aerosol profiles were not collected by UAVs for the decoupled cases presented in Table 2, the $\theta_e$ profiles and ceilometer measurements show evidence of boundary layer decoupling. These two decoupled cases have larger $\delta\sigma_{ext}$ than the coupled boundary layer cases in this study, leading to larger cloud-top $\delta RF$ as well. ACPM simulations were conducted using aerosol

concentrations based on the approximate average decoupled to coupled aerosol concentration ratio (50%) to estimate the difference in radiative flux. For the D05Sc case, simulations with 50% decreased cloud-base aerosol





concentrations show only slight differences δRF of 3 Wm$^{-2}$ and decreases in CDNC of 10%. The decrease in aerosol concentration resulted in increased supersaturation due to the low water uptake from fewer activating droplets. The increased supersaturation caused smaller aerosols to activate [*Raatikainen et al.*, 2013] and therefore, little change in

CDNC. The D06Cu was not influenced as much by low water uptake because the CDNC was much higher at 171 cm$^{-3}$ compared to 86 cm$^{-3}$ for D05Sc. The D06Cu the CDNC decreased by 42% and δRF decreased by 18 Wm$^{-2}$. Both decoupled cases still have a δRF greater than the coupled cases.

### 3.1.3 Satellite measurements closure

The satellite and simulated CDNC and S$_{max}$ measurements are presented in the bottom of Table 2. The method for

satellite retrieval of cloud properties could not be used for cases when cloud layers were too thin – which, unfortunately was the situation during the flights with the decoupled cloud layers. Nonetheless, Figure 4 shows the satellite image used to identify the clouds to calculate CDNC for C11Sc. Satellite retrieved cloud-base height and temperature are verified by ground-based ceilometer and temperature measurements. Figure 6 shows the top-down closures demonstrate that satellite-estimated CDNC and simulated CDNC are within a ± 30% expected concentrations, which

is limited by the retrieval of effective radius [*Rosenfeld et al.*, 2016]. The stratocumulus deck at the top of a well-mixed boundary layer (C11Sc) shows evidence of cloud-top inhomogeneous entrainment (see section 3.2). Freud et al. [2011] found that the inhomogeneous mixing assumption used to derive CDNC from satellite measurements resulted in an average over-estimate in CDNC of 30% (considering an adiabatic cloud droplet profile). Consequently, satellite-retrieved CDNC is reduced by 30% to account for the inhomogeneous entrainment assumption, which does

not necessarily reflect the actual magnitude of entrainment in the clouds. For example, in the C11Sc case, in-situ observations do indeed show cloud-top inhomogeneous entrainment; consequently, the usual 30% reduction in CDNC does not need to be applied (Table 2). Both stratocumulus cases (C11Sc, D05Sc) with cloud-top entrainment (Table 2) are similar to a case studied by Burnet and Brenguier [2007], in which cloud-top entrainment resulted in inhomogeneous mixing. In the following section, C11Sc and D05Sc are reanalyzed to include the effect of cloud-top

entrainment on simulated cloud properties using the inhomogeneous mixing assumption.

### 3.2 Entrainment

Based on the ground-based and UAV measurements, ACPM simulations over-estimate cloud radiative flux significantly for three cases (C11Sc, D05Sc, D06Cu). Section 3.1.2 identified that clouds in decoupled layers (D05Sc, D06Cu) have smaller radiative effects than predicted based on ground-based observations as aerosol (and CCN)

number concentrations in the decoupled layer are often smaller than in the surface layer. In this section, cloud-top entrainment is also shown to influence the radiative properties of two sub-adiabatic stratocumulus clouds, C11Sc and D05Sc.

The UAV observations show that both C11Sc and D05Sc have sub-adiabatic lapse rate measurements, compared to

simulated moist-adiabatic lapse rates within the cloud (Table 2). The difference between the observed and simulated lapse rates therefore suggests a source of heating in the cloud. The sub-adiabatic lapse rate is attributed to cloud-top entrainment by downward mixing of warmer air at cloud-top (e.g., Figure 7a).



Further evidence of cloud-top entrainment is shown through conserved variable mixing diagram analysis. In previous studies, a conserved variable mixing diagram analysis was used to show lateral or cloud-top entrainment by showing linear relationships between observations of conserved variables [*Burnet and Brenguier*, 2007; *Neggers et al.*, 2002; *Paluch*, 1979]. Paluch [1979] first observed a linear relationship of conservative properties (total water content, $q_t$ and liquid water potential temperature, $\theta_l$) between cumulus cloud cores and cloud edge, to show the cloud-free source of entrained air. Paluch [1979], Burnet and Brenguier [2007], Roberts et al. [2008], Lehmann et al. [2009] observed

decreases in CDNC and liquid water content in cumulus clouds as a function of distance from the cloud cores that indicate inhomogeneous mixing at the cloud edge. Burnet and Brenguier [2007] also show that $q_t$ is linearly proportional to liquid water potential temperature specifically for a stratocumulus cloud with cloud-top entrainment and inhomogeneous mixing. Direct observations of CDNC and liquid water content were not measured at Mace Head, so direct comparisons of CDNC and $q_t$ with Paluch [1979] and Burnet and Brenguier [2007] cannot be investigated

here. However, UAV measurements of cloud extinction (Eq. 1), which are related to CDNC ($CDNC = \int_0^\infty n(r)\,dr$) and liquid water content ($LWC = \int_0^\infty \frac{4}{3}\rho\pi r^3 n(r)\,dr$, $\rho$ is liquid water density), were measured and are found to be systematically lower than the adiabatic simulated cloud extinction (Figure 8).

    To apply the cloud-top mixing, a fraction of air at cloud-base and a fraction of air above cloud-top are mixed,

conserving $q_t$ and $\theta_e$. The fraction of air from cloud-base and cloud-top is determined with the measured equivalent potential temperature,

$$\theta_{e,c}(z) = \theta_{e,ent}X(z) + \theta_{e,CB}(1 - X(z)) \qquad (4)$$

where $\theta_{e,c}(z)$ is the equivalent potential temperature in cloud as a function of height, $\theta_{e,ent}$ is the equivalent potential temperature of the cloud-top entrained air, $\theta_{e,CB}$ is the equivalent potential temperature of air at cloud base, and $X(z)$

is the fraction of cloud-top entrained air as a function of height (referred to as the entrainment fraction). $\theta_{e,ent}$ $\theta_{e,c}(z)$ and $\theta_{e,CB}$ are measured parameters by the UAV and are not affected by latent heating from evaporation or condensation. The equivalent potential temperature, by definition, accounts for the total water content by including the latent heat released by condensing all the water vapor. Eq. (4) takes into account latent heating caused by evaporation of droplets. By rearranging Eq. (4), the entrained fraction is calculated as


$$X(z) = \frac{\theta_{e,c}(z) - \theta_{e,CB}}{\theta_{e,ent} - \theta_{e,CB}} \qquad (5)$$

    Figure 10a and b present the relationships between two conservative variables measured by the UAV (water vapor content, $q_v$, and $\theta_e$) for C11Sc and D05Sc. The $q_v$ is derived from relative humidity measurements and is equivalent to

the $q_t$ for sub-saturated, cloud-free air (i.e., < 100% RH).
    Figure 11 shows the relative humidity and $\theta_e$ profiles used in Figure 10. For both C11Sc and D05Sc, $\theta_{e,c}(z)$ is directly measured in-cloud, and $q_v$ and $\theta_e$ exhibit an approximately linear, proportional relationship (Figure 10; Eq. 4). The linear relationship is assumed to be a result of the cloud reaching a steady-state, with air coming from cloud-base and



cloud-top (e.g. cloud lifetime >> mixing time) . The observed in-cloud $q_v$ in Figure 10a and b is less than the

conservative variable $q_t$, however, the figure also includes $q_t$ based on simulated adiabatic and cloud-top entrainment conditions. Eq. (4) is used to derive the simulated cloud-top entrainment conditions (Figure 10a and b), where the fraction entrained is used to calculate $q_t$ and shows a linear relationship between $q_t$ and $\theta_e$. Measurements above cloud-top (RH < 95%) with $q_v > 5.1$ g kg$^{-1}$ and $q_v > 6.5$ g kg$^{-1}$ are used to represent the properties of the entrained air for C11Sc and D05Sc, respectively (Figure 10).


Figure 12 shows the sensitivity of the simulated cloud extinction profile, for the 11 August case, based on measurement uncertainties related to the entrained $q_v$ and $\theta$. The key variable for identifying the entrained fraction (Eq. 5), $\theta_{e,ent}$, is a function of $q_v$ and $\theta$, so a decrease in either parameter results in a proportional decrease in $\theta_{e,ent}$. Eq. (5) shows that entrainment fraction becomes more sensitive to the uncertainty related to the measurement of $\theta_e$ as the difference

between $\theta_{e,ent}$ and $\theta_{e,CB}$ approaches zero.

Table 3 shows $\delta\sigma_{ext}$, $\delta$RF, and CDNC for two cases with cloud-top entrainment (C11Sc and D05Sc) using two methods of accounting for the cloud top entrainment. One method applyies the entrainment fraction calculated in Eq. (5) and the other an entrainment parameterization, presented by Sanchez et al. [2016]. The entrainment parameterization

constrains the ACPM simulation to use the observed in-cloud lapse rate instead of assuming an adiabatic lapse rate. This is labeled the 'measured lapse rate' entrainment method in Table 3. In the sub-adiabatic cloud cases (C11Sc and D05Sc), the measured in-cloud lapse rate is lower than the adiabatic lapse rate, which leads to the condensation of less water vapor and subsequent activation of fewer droplets in the ACPM simulation. Similarly, the dryer and warmer entrained air (from above cloud-top) leads to evaporation of liquid water in the cloud. Previous observations of

stratocumulus cloud-top mixing suggest the entrainment is inhomogeneous [*Beals et al.*, 2015; *Burnet and Brenguier*, 2007], which implies that time scales of evaporation are much less than the time scales of mixing, such that a fraction of the droplets are evaporated completely and the remaining droplets are unaffected by the entrainment. The net decrease in CDNC subsequently results in less extinction of solar radiation compared to the purely adiabatic simulation.


The inclusion of inhomogeneous cloud-top entrainment improved the ACPM accuracy for both C11Sc and D05Sc using the measured lapse-rate and entrainment fraction methods (Figure 8, Table 3). After accounting for inhomogeneous entrainment, $\delta$RF decreased from 88 Wm$^{-2}$ to 47 Wm$^{-2}$ and 45 Wm$^{-2}$ to 14 Wm$^{-2}$ for D05Sc and D11Sc, respectively, using the entrainment fraction method. D05Sc simulations still yields significant $\delta$RF even after

accounting for inhomogeneous entrainment, likely because the cloud is in a decoupled BL, as noted in Section 3.1.2 to exhibit lower aerosol concentrations than those measured at the surface. The CDNC presented in Table 3 represents the CDNC at cloud base and did not change after applying the entrainment fraction method, however, the CDNC decreases with height for the entrainment fraction method rather than remain constant with height. Finally, the measured lapse rate entrainment method [Sanchez et al., 2016] does improve ACPM accuracy between in-situ and

satellite-retrieved cloud optical properties relative to the adiabatic simulations, but has greater $\delta\sigma_{ext}$ throughout the





cloud than the entrained fraction mixing method. For the measured lapse rate entrainment method δRF decreased from 88 Wm$^{-2}$ to 68 Wm$^{-2}$ and 45 Wm$^{-2}$ to 36 Wm$^{-2}$ for D05Sc and D11Sc respectively. The measured lapse rate entrainment parameterization resulted in lower δRF than the purely adiabatic simulations, however, δRF was minimized by directly accounting for the entrainment fraction.

**4 Conclusions**

This work presents measurements conducted in August 2015 at the Mace Head Research Station in Ireland, from multiple platforms including ground-based, airborne and satellites. As part of the BACCHUS (Impact of Biogenic versus Anthropogenic emissions on Clouds and Climate: towards a Holistic UnderStanding) European collaborative project, the goal of this study is to understand key processes affecting aerosol-cloud-radiative flux interactions. Seven
cases including cumulus and stratocumulus clouds were investigated to quantify aerosol-cloud interactions using ground-based and airborne measurements (bottom-up closure), as well as cloud microphysical and radiative properties using airborne measurements and satellite retrievals (top-down closure). An aerosol-cloud parcel model (ACPM) was used to link the ground-based, airborne and satellite observations, and to quantify uncertainties related to aerosols, cloud microphysical properties, and resulting cloud optical properties.


ACPM simulations represent bottom-up and top-down closures within uncertainties related to satellite retrievals for conditions with a coupled boundary layer and adiabatic cloud development. For these conditions the difference in radiative flux between simulations and *in-situ* observed parameters is less than 20 W m$^{-2}$. However, when entrainment and decoupling of the cloud layer occur, the ACPM simulations overestimate the cloud radiative flux. Of the seven
cases, two of the observed clouds occurred in a decoupled layer, resulting in differences in observed and simulated radiative flux (δRF) of 88 Wm$^{-2}$ and 74 Wm$^{-2}$ for the decoupled stratocumulus case on 5 August (D05Sc) and the decoupled cumulus case on 6 August (D06Cu) cases respectively. Adiabatic ACPM simulations resulted in a maximum cloud-top δRF value of 20 W m$^{-2}$ for coupled boundary layer cases and 74 W m$^{-2}$ for the decoupled boundary layer cases, after accounting for cloud-top entrainment. The reduction in aerosol concentrations in the
decoupled layer compared to ground-based measurements is a factor in overestimating decoupled cloud-top radiative flux with the ACPM, however simulations with 50% decreased aerosol concentrations show only slight differences δRF of 3 W m$^{-2}$ and decreases in CDNC of 10% for D05Sc. For D06Cu δRF decreased by 18 Wm$^{-2}$ and the CDNC decreased by 42%. Even after decreasing the aerosol concentration by 50% both decoupled cases have δRF values significantly higher than the coupled boundary layer cases ($< 20$ W m$^{-2}$).


For the cases with cloud-top entrainment, D05Sc and the coupled stratocumulus case on 11 August (C11Sc), liquid water content is one of the major factors in overestimating cloud-top radiative flux with the ACPM. For these cases, the measured in-cloud lapse rates are lower than adiabatic lapse rates, suggesting a source of heat due to entrainment of warmer, drier air from above the cloud. Furthermore, linear relationships between conservative variables, simulated total water vapor, $q_t$, and equivalent potential temperature, $\theta_e$, also suggest mixing between air at cloud-base and cloud-
top. For D05Sc, after accounting for cloud top entrainment by applying the entrainment fraction δRF decreased from



88 W m$^{-2}$ to 46 Wm$^{-2}$. For the coupled boundary layer case with entrainment (C11Sc) the δRF decreases from 45 Wm$^{-2}$ to 14 Wm$^{-2}$ after accounting for cloud top entrainment with the entrainment fraction.

Based on airborne observations with UAVs, cloud-top entrainment is only observed on 2 out of the 13 flight days and decoupling of the boundary layer occurs on 4 of the 13 flight days. These cases illustrate the need for *in-situ* observations to quantify entrainment mixing and cloud base CCN concentrations particularly when the mixing state of the atmosphere is not known. Even greater discrepancies between the surface and decoupled layer CCN concentrations will occur in the presence marine biogenic sources such as tidal regions and local anthropogenic [*Colin*
*D. O'Dowd*, 2002]. Using ground-based observations to model clouds in decoupled boundary layers and not including cloud top entrainment are shown to cause significant differences between observations and simulation radiative forcing and therefore, should be included in large scale modeling studies to accurately predict future climate forcing.

UAV measurements were coordinated with 13 days of satellite overpasses and cloud microphysical properties were
retrieved for four of the cases. When accounting for entrainment, the differences between simulated and satellite-retrieved CDNC are within the expected 30% accuracy of the satellite retrievals [*Rosenfeld et al.*, 2016]. However, in-situ measurements are necessary to refine satellite retrievals to allow cloud properties to be studied on larger spatial scales.

**Acknowledgements.** The research leading to these results received funding from the European Union's Seventh
Framework Programme (FP7/2007-2013) project BACCHUS under grant agreement n°603445. EU H2020 project ACTRIS-2 under the grant agreement No. 654109 is also acknowledged for supporting the Mace Head Research Station. K.A. Nicoll acknowledges a NERC Independent Research Fellowship (NE/L011514/1). D. Ceburnis acknowledges the Irish EPA (2012-CCRP-FS.12). J. Preissler acknowledges the Irish EPA (2015-CCRP-FS.24). R. Calmer acknowledges financial support from Meteo France. K. J. Sanchez acknowledges the Chateaubriand
Fellowship. The authors also acknowledge Kirsten Fossum for the collection of SMPS data.

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





**Table 1. UAV research flights conducted at Mace Head, Ireland and measured parameters in 2015. Flight start and end times are in UTC. Suomi NASA Polar-orbiting Partnership satellite overpasses occurred at approximately 13:00 UTC. Measurements include relative humidity (RH), temperature (T), pressure (P), 3-dimensional wind vectors (3D Winds), optical particle counter (OPC) and cloud sensor measurements of cloud droplet extinction.**

| Date | Flight | Start Time | End Time | RH | T | P | 3D Winds | OPC | Cloud |
|------|--------|-----------|----------|----|---|---|----------|-----|-------|
| 30-Jul | 4 | 12:41 | 13:19 | x | x | x | | x | |
| 30-Jul | 5 | 14:00 | 14:44 | x | x | x | | | x |
| 30-Jul | 6 | 16:04 | 16:42 | x | x | x | | x | |
| 01-Aug | 7 | 11:30 | 12:13 | x | x | x | | x | |
| 01-Aug | 8 | 12:35 | 13:16 | x | x | x | | | x |
| 01-Aug | 9 | 14:00 | 15:20 | x | x | x | x | | |
| 01-Aug | 10 | 15:54 | 16:43 | x | x | x | | x | |
| 05-Aug | 11 | 11:47 | 12:29 | x | x | x | | | x |
| 05-Aug | 13 | 13:36 | 14:26 | x | x | x | x | | |
| 05-Aug | 14 | 14:42 | 15:29 | x | x | x | | | x |
| 06-Aug | 16 | 11:55 | 12:37 | x | x | x | | | x |
| 06-Aug | 17 | 13:51 | 15:16 | x | x | x | x | | |
| 10-Aug | 19 | 13:41 | 14:10 | x | x | x | | | x |
| 10-Aug | 20 | 14:42 | 15:45 | x | x | x | x | | |
| 10-Aug | 21 | 16:00 | 16:45 | x | x | x | | | x |
| 11-Aug | 23 | 12:00 | 12:47 | x | x | x | | | x |
| 11-Aug | 24 | 13:11 | 14:05 | | x | x | x | | |
| 11-Aug | 25 | 14:25 | 15:10 | x | x | x | | | x |
| 11-Aug | 26 | 15:29 | 16:22 | | x | x | x | | |
| 11-Aug | 27 | 16:58 | 17:33 | | x | x | | x | |
| 15-Aug | 29 | 12:19 | 13:03 | x | x | x | | x | |
| 15-Aug | 30 | 13:46 | 14:31 | | x | x | x | | |
| 15-Aug | 31 | 15:08 | 16:14 | x | x | x | | | x |
| 16-Aug | 32 | 12:30 | 13:20 | x | x | x | | x | |
| 16-Aug | 33 | 13:40 | 14:00 | x | x | x | | x | |
| 17-Aug | 34 | 11:30 | 12:24 | x | x | x | | | x |
| 17-Aug | 35 | 13:45 | 14:34 | x | x | x | | x | |
| 21-Aug | 36 | 12:21 | 13:12 | | x | x | | x | |
| 21-Aug | 37 | 13:40 | 14:25 | x | x | x | | | x |
| 21-Aug | 38 | 15:17 | 16:26 | x | x | x | x | | |
| 21-Aug | 39 | 16:53 | 17:27 | x | x | x | | | x |
| 22-Aug | 40 | 9:29 | 10:12 | x | x | x | | | x |
| 22-Aug | 41 | 10:47 | 11:37 | x | x | x | | x | |
| 22-Aug | 42 | 12:52 | 13:53 | x | x | x | x | | |
| 22-Aug | 43 | 14:22 | 14:59 | x | x | x | | x | |
| 27-Aug | 45 | 10:21 | 11:10 | x | x | x | | x | |
| 27-Aug | 46 | 11:27 | 12:13 | x | x | x | | | x |
| 27-Aug | 47 | 13:11 | 13:45 | | | x | | | x |
| 27-Aug | 48 | 15:09 | 15:23 | x | x | x | x | | |
| 27-Aug | 49 | 17:20 | 17:50 | x | x | x | | x | |
| 28-Aug | 50 | 14:25 | 14:49 | x | x | x | | x | |




**Table 2.** UAV observations of cloud heights and temperatures and cloud property estimates based on ground measurements. Ground-based Hoppel minimum diameter ($D_{min}$) is used to estimate CDNC. ACPM simulation and satellite results are also presented, as well as differences between simulated and observation-derived cloud-top extinction and cloud-top radiative flux. Case abbreviations include if they are coupled (C) or decoupled (D), the day of the month and cloud types, cumulus (Cu) or stratocumulus (Sc).

| | Coupled BL | | | | | Decoupled BL | |
|---|---|---|---|---|---|---|---|
| | 01Aug Cumulus (C01Cu)[a] | 05 Aug Cumulus (C05Cu) | 10 Aug[b] Cumulus (C10Cu) | 11 Aug[c] StratoCu (C11Sc) | 21 Aug Cumulus (C21Cu) | 05 Aug[c] StratoCu (D05Sc) | 06 Aug Cumulus (D06Cu) |
| *In-situ* **Ground-based and UAV Measurements** | | | | | | | |
| **Cloud-base height (m)** | 800 | 430 | 650 | 1200 | 460 | 1490 | 2180 |
| **Cloud-base temperature (ºC)** | 7.4 ±0.1 | 10.6 ±0.2 | 8.1 ±0.1 | 3.7 ±0.1 | 10.4 ±0.1 | 6.5 ±0.2 | -2.1 ±0.2 |
| **Cloud-top height (m)** | 1040 | 710 | 1720 | 1460 | 960 | 1630 | 2400 |
| **Cloud-top temperature (ºC)** | 5.7 ±0.1 | 8.7 ±0.2 | 1.8 ±0.1 | 2.4 ±0.2 | 7.6 ±0.1 | 5.8 ±0.2 | -3.1 ±0.4 |
| **Measured lapse rate in-cloud (K km$^{-1}$)** | 5.7 | 6.1 | 5.1 | 4.7 | 6.0 | 4.1 | 6.3 |
| **Number of cloud layers** | 1 | 2[f] | 1 | 1 | 1 | 2[g] | 2[g] |
| **Hoppel $D_{min}$ (nm)** | 74 ±6 | 78 ±16 | 73 ±8 | 83 ±7 | 83 ±5 | 78 ±16 | 80 ±9 |
| **Hoppel $D_{min}$ CDNC (> Hoppel $D_{min}$, cm$^{-3}$)** | 129 ±5 | 69 ±8 | 105 ±11 | 87 ±5 | 94 ±12 | 69 ±8 | 164 ±13 |
| **Hoppel minimum critical supersaturation ($S_{crit}$)** | 0.43 ±0.03 | 0.61 ±0.10 | 0.37 ±0.11 | 0.37 ±0.05 | 0.41 ±0.10 | 0.61 ±0.10 | 0.31 ±0.06 |
| **ACPM Simulation and Satellite-derived Cloud Properties[d]** | | | | | | | |
| **Simulated moist adiabatic lapse rate (K km$^{-1}$)** | 5.0 | 4.5 | 4.9 | 5.7 | 4.5 | 5.1 | 6.4 |
| **Simulated Cloud-top droplet $r_e$ (µm)** | 10.3 ±0.1 | 14.4 ±0.3 | - | 11.3 ±0.2 | 14.2 ±0.4 | 10.0 ±0.1 | 8.2 ±0.2 |
| **Cloud-top extinction difference ($\delta\sigma_{ext}$, km$^{-1}$)** | - | 11 ±25 | - | 36 ±12 | 52 ±42 | 37 ±6 | 34 ±7 |
| **Cloud-top radiative flux difference ($\delta$RF, W m$^{-2}$)[e]** | - | 11 ±26 | - | 45 ±11 | 20 ±6 | 88 ±8 | 74 ±12 |
| **Simulated CDNC (cm$^{-3}$)** | 135 ±16 | 60 ±12 | 105 ±18 | 88 ±12 | 105 ±31 | 86 ±10 | 171 ±17 |
| **Satellite estimated CDNC (cm$^{-3}$)** | 109 | - | 85 | 58 (83)[h] | 104 | - | - |
| **Simulated $S_{max}$ (%)** | 0.45 ±0.09 | 0.45 ±0.18 | 0.36 ±0.15 | 0.36 ±0.09 | 0.40 ±0.20 | 0.76 ±0.04 | 0.33 ±0.06 |
| **Satellite estimated $S_{max}$ (%)** | 0.34 | - | 0.27 | 0.48 | 0.34 | - | - |

[a] C/D – coupled / decoupled; xx – date in August 2015; Sc / Cu – stratocumulus / cumulus cloud





[b] Precipitation occurred on 10 Aug
[c] Accounting for entrainment improves model / measurement closure (Table 2)
[d] The error includes the potential error of ±20% in updraft velocity and the standard error of the CCN concentration measurements.
[e] The difference between the observed (calculated from UAV extinction measurements) and simulated radiative flux. The error includes the potential error of ±20% in updraft velocity and the standard

error of the CCN concentration measurements.
[f] The measurements and results in this column represent the lower of the two clouds.
[g] Altitude of top cloud level that is used to calculate cloud radiative flux.
[h] Excluding the correction for the inhomogeneous entrainment assumption in parentheses



**Table 3. Results of the application of entrainment fraction and the measured lapse rate entrainment parameterization for two clouds with observed cloud-top entrainment.**

| | Coupled BL (C11Sc) | | Decoupled BL (D05Sc) | |
|---|---|---|---|---|
| **Entrainment method** | Entrainment Fraction | Measured Lapse Rate | Entrainment Fraction | Measured Lapse Rate |
| **Cloud-top extinction difference ($\delta\sigma_{ext}$, km$^{-1}$)** | 14 ±10 | 26 ±11 | 16 ±5 | 26 ±6 |
| **Cloud-top radiative flux difference ($\delta RF$, W m$^{-2}$)[a]** | 14 ±9 | 36 ±11 | 47 ±8 | 68 ±13 |
| **Cloud base simulated CDNC[b]** | 88 ±12 | 83 ±12 | 86 ±10 | 68 ±10 |

[a] The difference between the observed (calculated from UAV extinction measurements) and simulated radiative flux. The error includes the potential error of ±20% in updraft velocity and the standard error of the CCN concentration measurements.

[b] The simulated CDNC is unchanged at the cloud base for the entrainment fraction method, however the CDNC decreases with height.





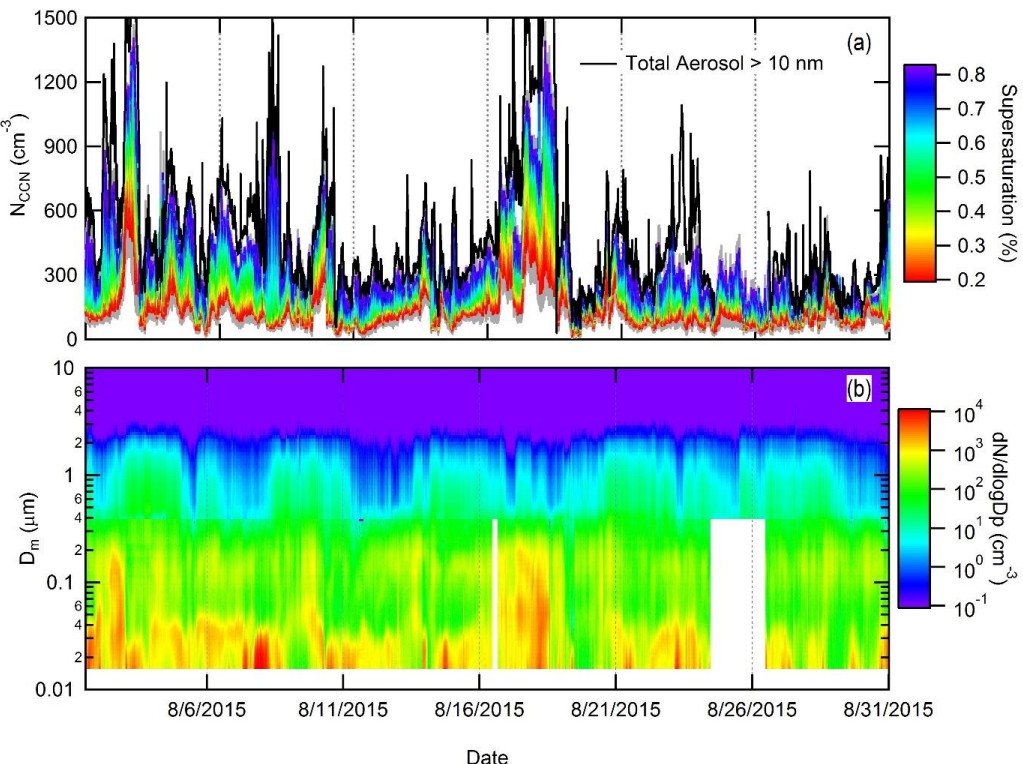

**Figure 1. Time series for the month of August 2015 at Mace Head Ireland of ground-based CCN concentrations (top) and merged SMPS and APS number size distributions (bottom).**






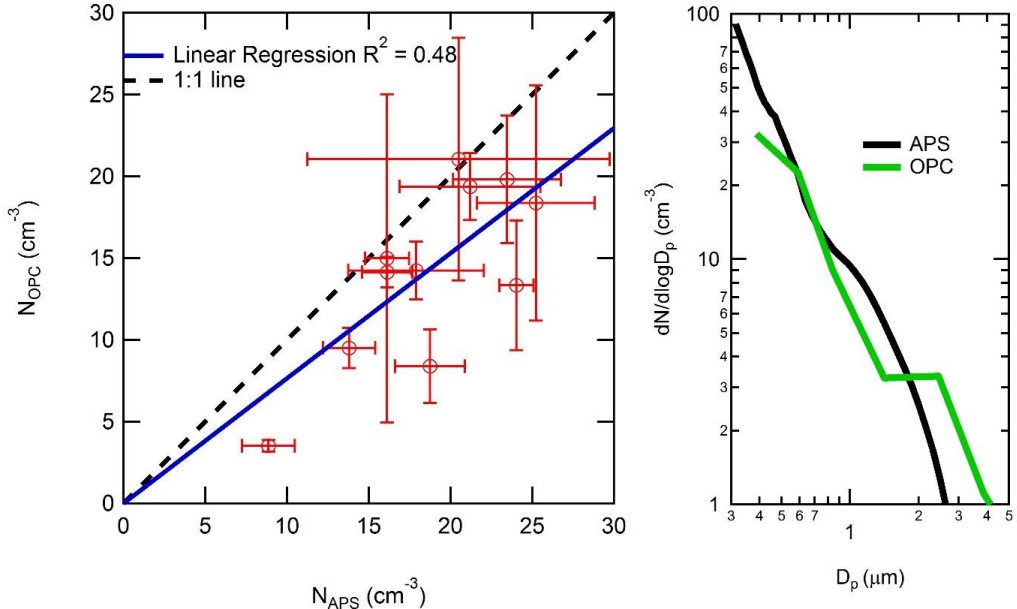

**Figure 2. OPC concentrations with particle diameters (Dp) greater than 0.5 um (left) from 11 UAV research flights, listed in Table 1, plotted against APS concentrations (Dp > 0.5 um) at Mace Head Research Station (red circles). Error bars represent ±1 standard deviation. The points are fit with a linear regression (blue line). OPC data was averaged between 40 and 80 m asl. Averaged OPC and APS number size distributions averaged for the 11 flights (right).**






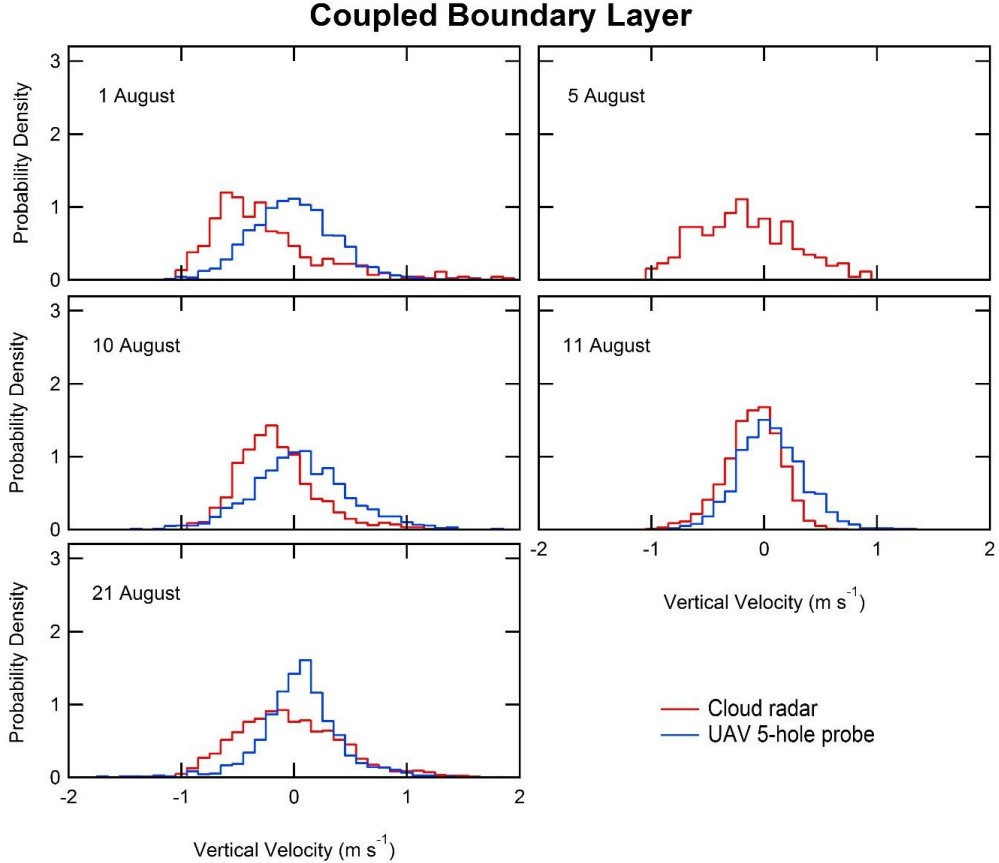

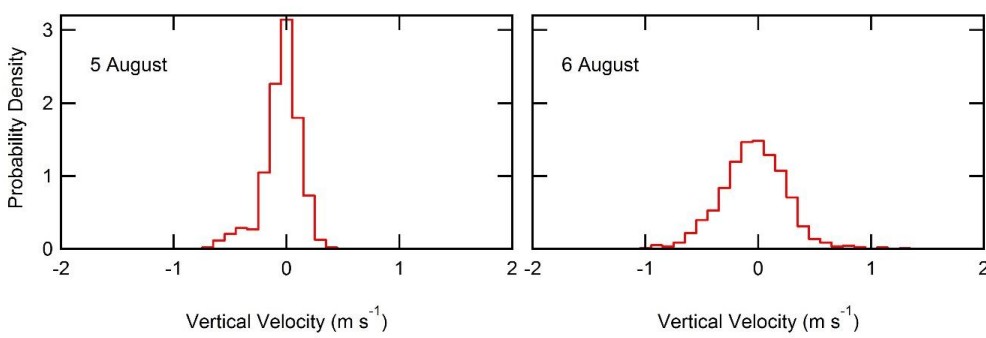

**Figure 3. Normalized observed vertical velocity distributions measured by the cloud radar and UAV for each case presented in Table 2.**





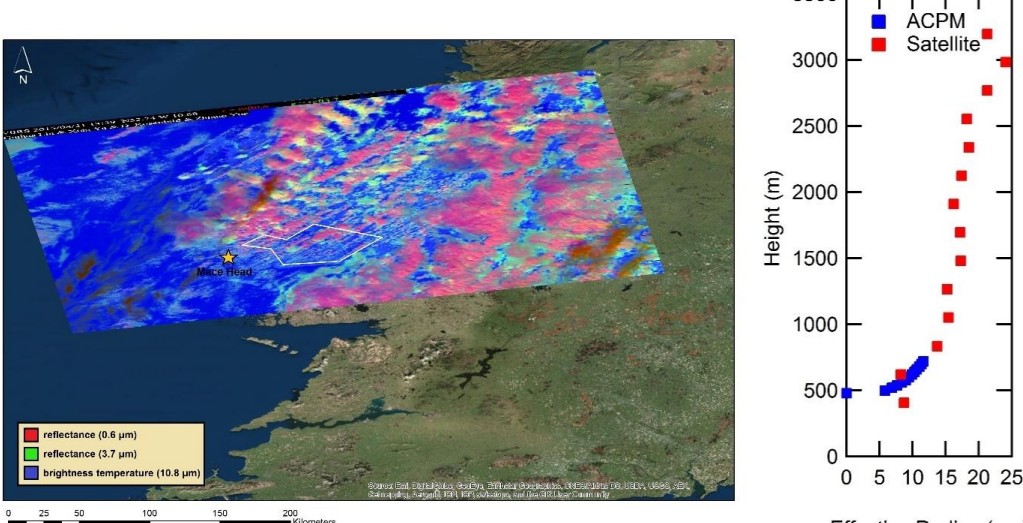

**Figure 4. Suomi NPP satellite RGB composite image for 21 August 2015 (left). Mace Head Research Station and UAV flight location are indicated by the yellow star. The white polygon represents the zone for retrieving cloud properties – which is represented by the profile of cloud effective radius (right). Effective radius profiles are presented for both the Suomi NPP satellite (red) and the ACPM (blue).**






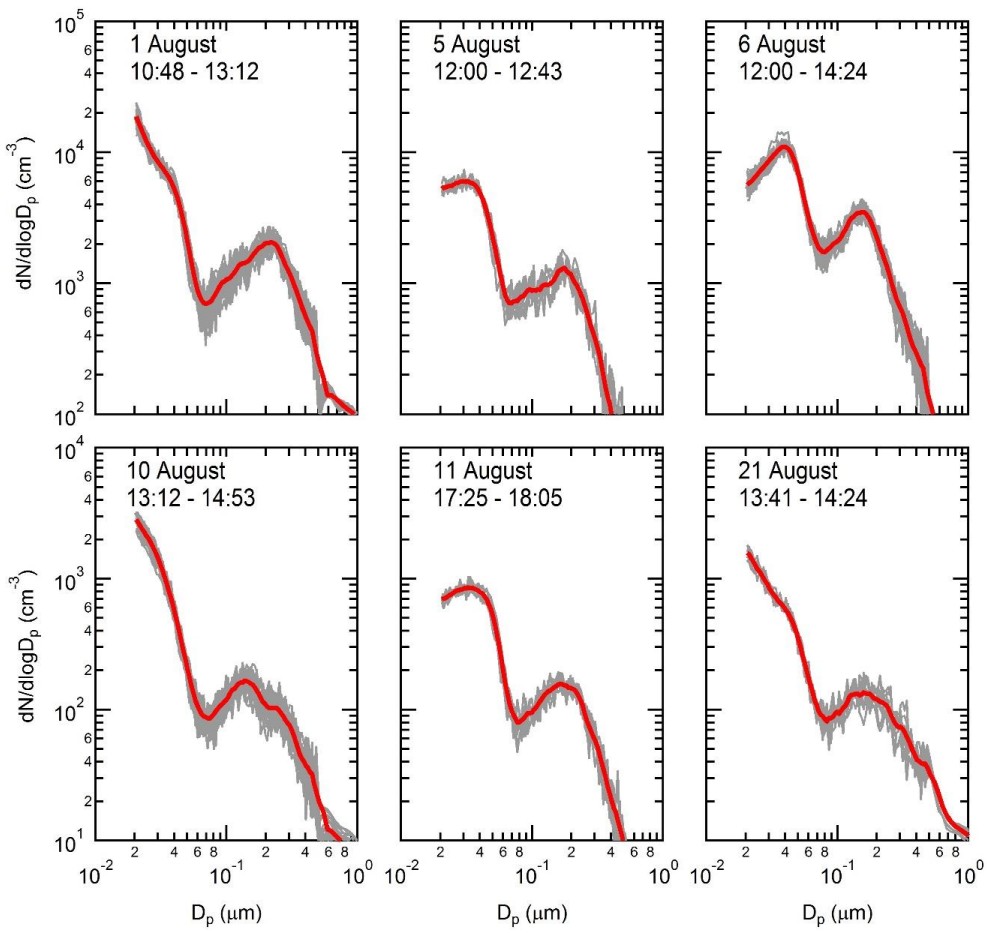

**Figure 5. SMPS and APS derived size distributions used for each case study in Table 2. The 5 August size distribution is used for both the coupled and decoupled case. Individual distributions (grey) are from the indicated time ranges in the figure. The time ranges are in UTC. Average distributions are shown in red.**





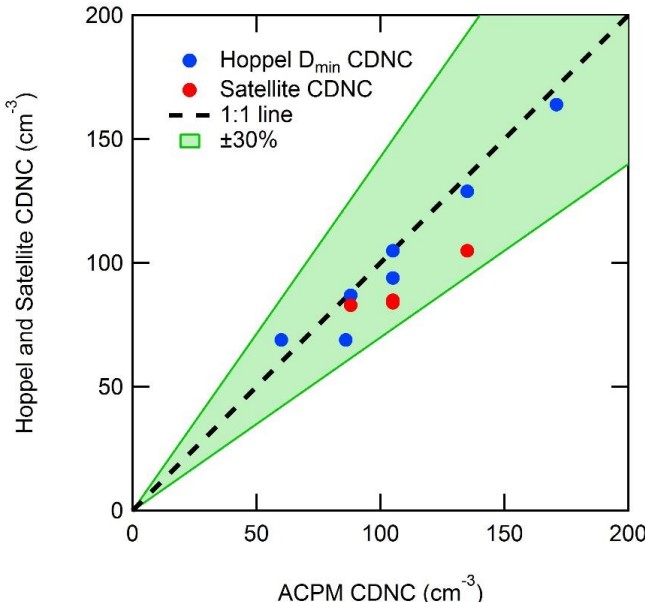

**Figure 6. Comparison of simulated CDNC from ACPM with both Hoppel minimum diameter ($D_{min}$) derived CDNC (blue) and satellite estimated CDNC (red). CDNC plotted are from the listed cloud cases in Table 2. The green shaded region represents Hoppel and Satellite CDNCs within 30% of ACPM simulation CDNC.**





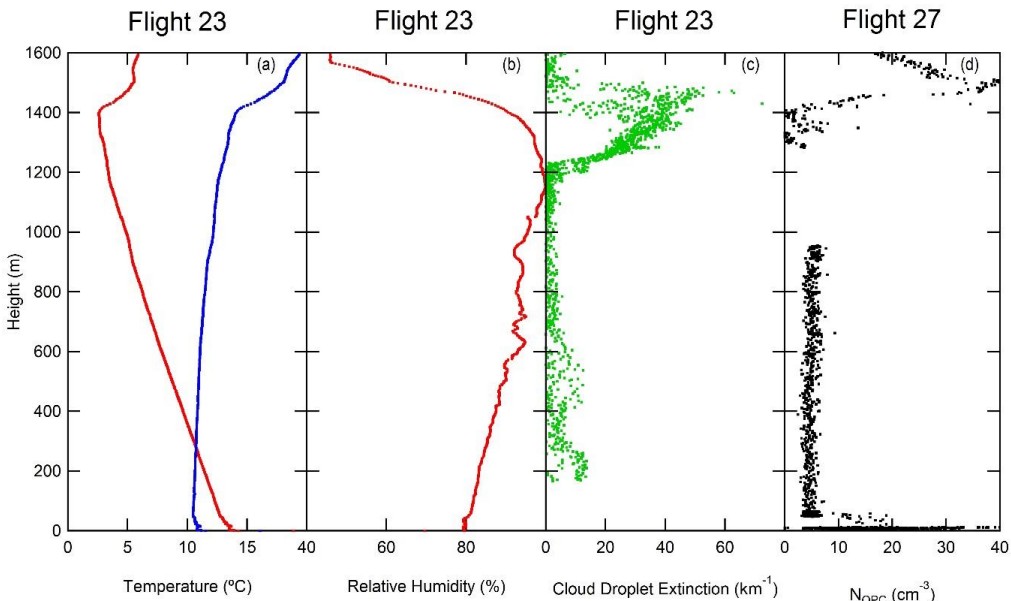

**Figure 7. Vertical profiles of temperature, virtual potential temperature ($\theta_v$), relative humidity, cloud droplet extinction and OPC total aerosol concentration. The figure consists of measurements collected from flights 23 and 27 on 11 August 2015 between 12:00 - 12:47 and 16:58 -17:33 respectively. The cloud level is between 1200 m to 1480 m in flight 23, and lowered to approximately 980 m to 1280 m in flight 27. OPC measurements that occurred in the cloud have been removed.**





**Figure 8. Vertical profiles of measured and simulated cloud extinction from flights D05Sc, C11Sc and C21Cu (left figures; Table 2).** *In-situ* **measurements are classified into cloud, cloud-transition and cloud-free observations. The difference between UAV-observed (green measurements) and ACPM-simulated cloud extinction (black line) on left figures are used to calculate ($\delta\sigma_{ext}$) as a function of altitude in the right-hand side figures. The slope of the best fit through in-cloud measurements (red line) represents the increase in $\delta\sigma_{ext}$ as a function of cloud thickness.**





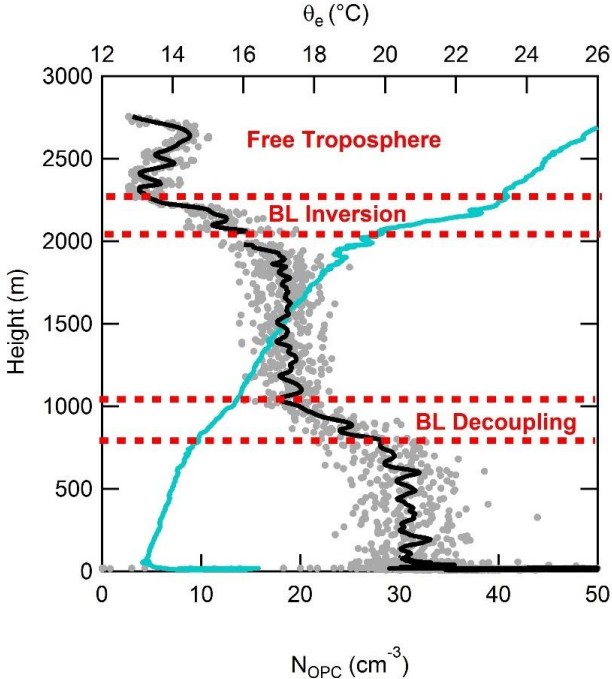

**Figure 9. Flight 10 UAV vertical profile of OPC aerosol number concentrations (Dp > 0.5 um) (grey) with a 20 second running mean (black) and equivalent potential temperature ($\theta_e$, light blue) illustrate decoupling of the boundary layer. In-cloud OPC measurements (2000 m- 2050 m) have been removed.**




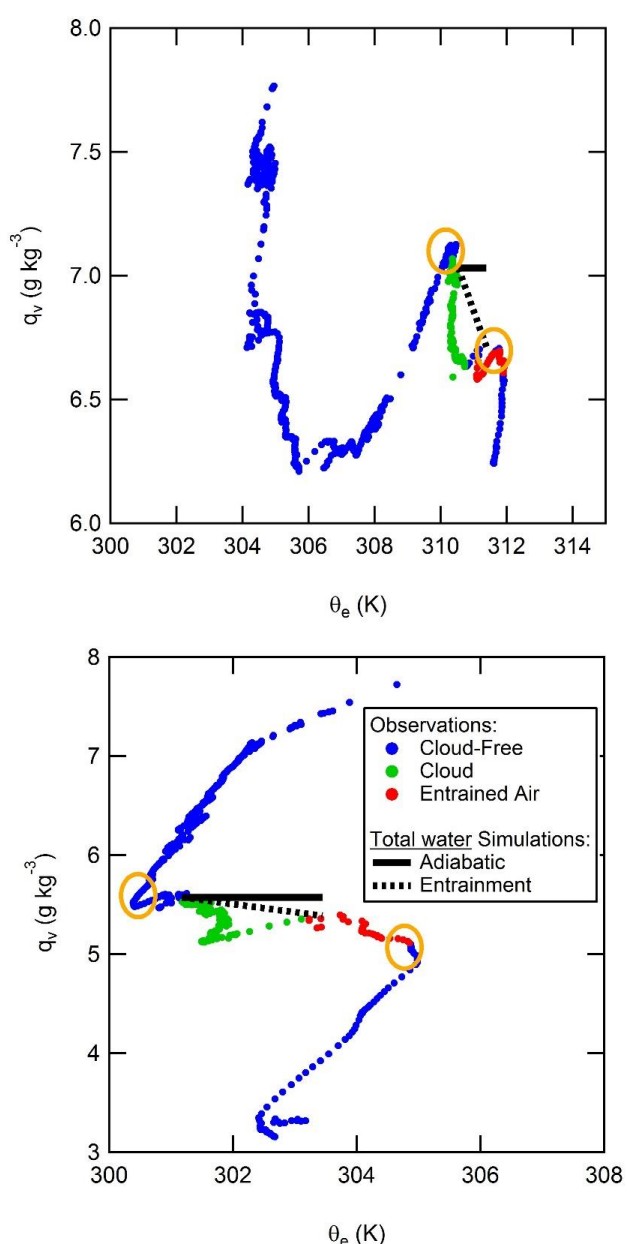

**Figure 10. Conservative variables, water vapor content ($q_v$, conservative in subsaturated conditions and derived from RH measurements) and equivalent potential temperature ($\theta_e$) identify mixing between cloud air and entrained air. Measurements are defined as cloud-free (blue), in-cloud (green) or entrained air sources (red). The orange circles highlight what is suggested to be the non-mixed sources of air.**




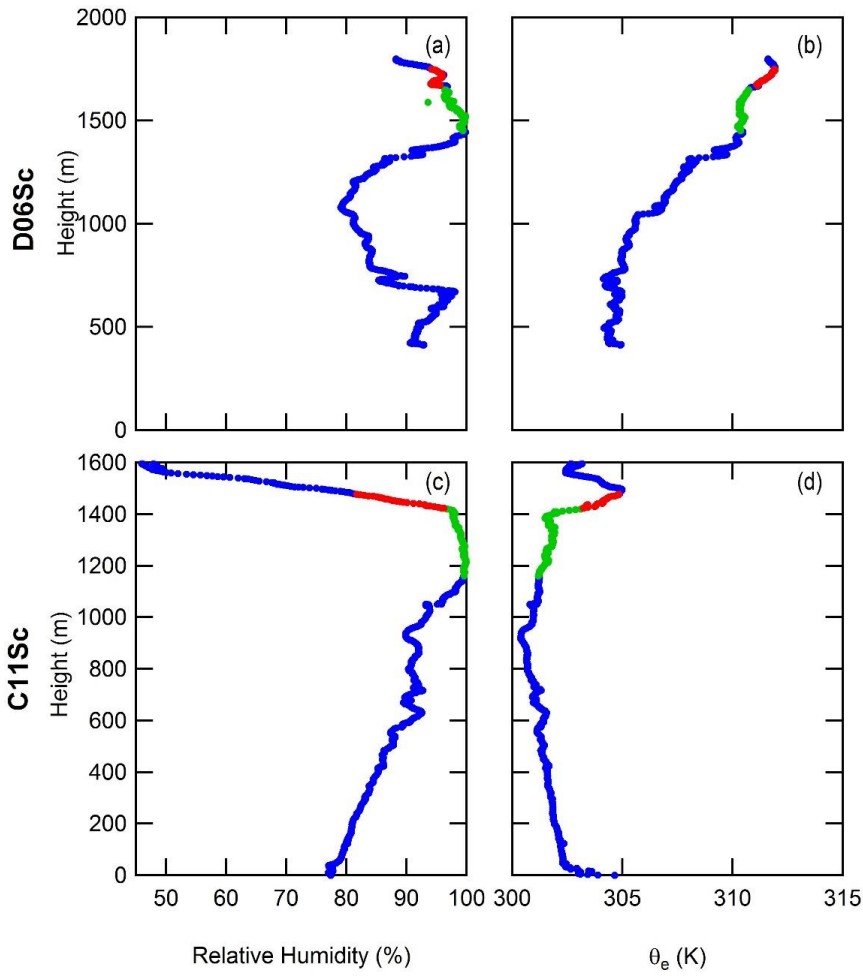

**Figure 11. UAV vertical profiles of relative humidity (a, c) and $\theta_e$ (b, d) for flights D06Sc and C11Sc, used in Figure 9. Profiles are defined as cloud-free (blue), in-cloud (green) or entrained air sources (red).**





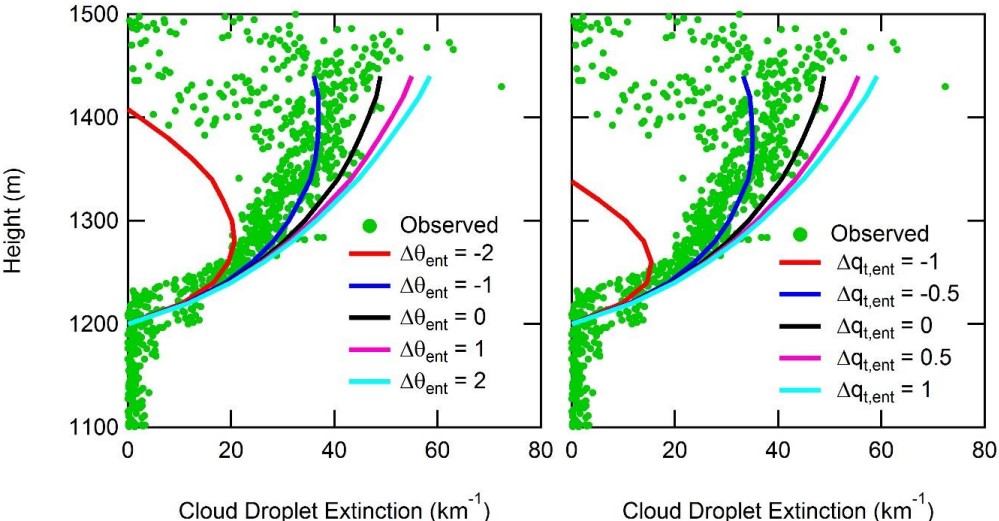

**Figure 12. Sensitivity of simulated cloud extinction based on variability of entrained air potential temperature ($\Delta\theta_{ent}$) and entrained air total water mixing ratio ($\Delta q_{t,ent}$) for the C11Sc case. Black lines are equivalent to the adiabatic simulation with entrainment from Figure 7c.**