# Peer review of "Top-down and Bottom-up aerosol-cloud-closure: towards understanding sources of uncertainty in deriving cloud radiative flux"

_Atmospheric Chemistry and Physics, 2017_

## Referee Comment (RC1) · Anonymous Referee #1 · 12 May 2017

The authors provide an analysis of cloud droplet closure using data collected at Mace Head, Ireland during summer 2015. The dataset includes surface based aerosol and remote sensing data from the Mace Head station. In addition, in situ vertical profile data was collected from a new UAV platform, which was deployed with a rotating payload comprising of meteorological probes, an aerosol optical sizing spectrometer and a cloud extinction monitor. Finally, the authors also make use of satellite cloud remote sensing products.

The authors conduct an aerosol-cloud microphysical closure analysis from the surface based data input into a parcel model (bottom-up) and from the satellite and in situ cloud extinction (top-down) to assess the uncertainty in deriving shortwave cloud radiative effects associated with microphysics. The authors find that when they account for reductions in cloud drop number concentration associated with entrainment, the difference between modelled and observed shortwave fluxes are reduced. The authors also find that decoupled clouds result in larger differences between modelled and observed shortwave fluxes, compared to well mixed cases.

Overall the paper is interesting and suitable for publication in ACP. I have a number of minor points listed below, which I urge the authors to consider before the paper is finalized.

General point: It might be useful to clarify in the abstract (and in sections before you define RF) that you are discussing shortwave radiative flux

L80 "surface latent heat flux, (i.e. evaporative cooling at the surface)" – this is misleading: surface latent heat flux does not induce cooling. It is independent of the heat budget at the surface. The mechanism, described in BW97, results in decoupling because under high LHF, there is a larger jump in buoyancy flux at cloud base, the cloud layer drives the turbulent motions and a zone of negative buoyancy flux develops in the sub-cloud layer. When this zone becomes too large it becomes dynamically favorable for the cloud layer to decouple from the sub-cloud layer.

L81 BW97 claim that drizzle is not necessary for their "deepening-warming decoupling" mechanism, however they do show that it can have a substantial impact on the promotion of negative sub-cloud buoyancy fluxes and induce decoupling.

L87 also related, moving air over a higher SST does not induce cooling. Suggest reviewing Stevens, 2002, Bretherton and Wyant, 1997 and Schubert et al., 1979 (not exhaustive list) for information about the mechanism of decoupling driven by increased surface latent heat fluxes and negative sub-cloud buoyancy fluxes.

L123 (sp) Nafion.

L145-147 how is the scaling done? In Figure 7 and Figure 11, RH values are shown to

be <100% in the cloud layer.

L155 typo - Aerosols

L204-206 Mixing state: can you clarify what you mean by "externally mixed types of particles". You then state that aerosols are internally mixed: is it fair to say that aerosols are internally mixed when this paper is discussing evidence of a significant fraction of air entrained into the boundary layer from above? Would aerosols from the free troposphere not have different chemical characteristics from the boundary layer? The phrase "lack of aerosol sources" is also ambiguous.

L215-226 Does the model include the effects of coalescence scavenging, which may be quite significant for a marine cloud over the 2-hour period given here.

L222 should there be a negative sign in your equation for the adiabatic cooling term (i.e. –gdz/cp)?

L340-342 I think you could be a bit clearer about how you come to this conclusion from the data shown in Table 2.

L374-390 in both well-mixed and decoupled boundary layers, there are diabatic processes affecting the cloud layer namely, long-wave cooling of the cloud top, short wave absorption, drying due to drop sedimentation. To what extent do these processes interfere with the assumption of a cloud parcel being a mixture of cloud base air and entrained air?

Fig 10: suggest putting the flight details in the caption (like Fig 11) for clarity

L388-400 I think this section could be reworded to improve its clarity. I also have a few concerns: 1) It's not clear what you are referring to with the linear proportional relationship (L392). As you clarified, the qv=qt is only true outside the cloud, but if this mixing diagram is only used to illustrate processes in the cloud, what new information do you get for cloudy air with the addition of the second dimension (qv) over the 1D theta-E mixing calculation done with Eq.4? 2) The dashed line is linear by design, on

a qt axis. Since qt=qv at the two end points these would indeed be the end points of the dashed line but on this qv axis the line would be curved 3) It is not clear what the adiabatic line is supposed to represent. Why does theta-E change during an adiabatic process?

L417 what is the sensitivity of cloud extinction if mixing is homogeneous v.s. inhomogeneous compared to, say, the magnitude of the entrainment? Are there any other clues from your data set that could help confirm that the inhomogeneous process is a suitable assumption?

L470 What was happening on the other cases? Was the cloud layer more vigorously mixed, such that entrainment warming and drying was homogenized through the layer more rapidly?

L474 "presence (of) marine biogenic..."

L474 local anthropogenic...what?

L475 "observations and simulat(ed)"?

---

## Referee Comment (RC2) · Anonymous Referee #2 · 23 May 2017

Summary: This manuscript presents an observational analysis to understand sources of uncertainty in deriving cloud radiative flux. The observations are from a number of platforms, including ground based, UAV, and satellite measurements. They used a 1-D microphysical model in conjunction with observations to derive microphysical and optical properties of observed clouds. The differences were found in radiative fluxes between the simulated and the observed. They concluded that the cloud-top entrainment is an important source of uncertainty for the cloud radiative flux calculation; it is particularly true for decoupled cloud boundary layers because ground-based measurements are no longer enough to obtain reliable data in the decoupled cloud layer. Authors' overall analysis technique is good and their conclusion is important and in-

teresting. My main criticism is that some discussions and figures are not clear and confusing. I recommend publication after following comments are addressed.

I am wondering about the significance of showing the cloud-top extinction in Table 2 and 3. Even though the cloud-top radiative flux differences (Delta FR) in the two decoupled cases are larger than those in the coupled cases, delta sigma_ext values are similar for all the cases as shown in Table 3. The cloud-top value delta sigma_ext doesn't seem to mean a lot in terms of cloud optical property. Because the cloud-top radiative flux (RF) depends on the optical depth as shown in (2), it is probably more appropriate to show cloud optical depth (tau).

Page 2, line 71: "Such decoupled layers often contain two distinct cloud layers, . . . a lower layer within the well-mixed surface layer and a higher decoupled residual layer between the free atmosphere and surface layer". I don't think the surface layer can be well mixed because turbulent eddies there are too small near the surface to produce strong mixing. You probably meant surface based mixed layer. That is, a mixed layer that is connected to, but deeper than the surface layer. Why do you call a decoupled layer "residual layer"? Is there turbulence source in the decoupled layer? Does it have clouds?

Page 3, line 75: "the surface mixed layer". Surface based mixed layer?

Page 3, line 77 and line 80: " . . . involve cloud heating and surface cooling" and " i.e., evaporative cooling at the surface" I am not sure what is meat by the "surface cooling" or "evaporative cooling". Note that the surface evaporative cooling by surface moisture flux only cools the ocean surface, not the sub-cloud layer. I do not think the "surface evaporative cooling" directly contributes to the decoupling. Could you give a bit more explanation on this? An increase in the moisture flux with increasing SST enhances the cloud layer buoyancy flux, which intensifies the cloud-top entrainment to mix warmer and drier air into clouds, leading to negative buoyancy flux below cloud base.

Page 8, line 281-282 about Figure 8. Could you put the flight code (D05Sc, C11Sc,

and C21Cu) inside the plot boxes? That would be easy to see. The caption of Figure 8 mentions the difference between UAV-observed (green measurements) and ACPM-simulated (black line) to calculate delta sigma_ext. But it looks like you also calculate the cloud free values too. Although the (a)-(f) are labeled in each plot, they are not used in the caption.

Page 10, line 354-357: "The UAV observations show both C11Sc have sub-adiabatic lapse rate measurements, compared to simulated moist-adiabatic lapse rates within the cloud (Table 2). . . . The sub-adiabatic lapse rate is attributed to cloud-top entrainment . . .. at cloud-top (e.g., Figure 7a)" Where is the comparison between the observed and simulated lapse rate? I only see the simulated values in Table 2. Could you draw a line in Figure 7a to show the adiabatic lapse rate? It is hard to see the lapse rate is sub-adiabatic.

Page 11-12, 391-399: "For both C11Sc and D05SC,. . ... . exhibit an approximately linear, proportional relationship (Figure10; Eq. 4.) . . . ". This paragraph is a bit confusing. What flights do those curves come from in Fig. 10? Could you state clearly which part you were referring to that is linear? In Fig. 10, the cloudy part (green curve) is not linear because $q_v$ is not conserved variable for condensation/evaporation process.

What is meant by "entrained air"? Does it consist of both free air and turbulent air or only free atmosphere and non-mixed air? Does it contain any cloud droplets? If not, why is it (red curve) not linear, particularly for the top panel plot?

What is the flight code (or number) for these two plots in Fig. 10? Please identify the blue dashed line in the text when discussing the entrainment conditions. There is no (a) and (b) in Fig 10. "Measurements above cloud-top (RH < 95%) with $q_v$ > 5.1 g kg-1 and $q_v$ > 6.5 g kg-1 are used to represent the properties of the entrained air". How do you choose this criterion for the entrained air? You should specify clearly the properties of the non-mixed sources of air: what are the values of theta_e and q_v of the air source? The orange circles include too many possibilities of these values.

Line 391: "Figure 11 shows the relative humidity and theta_e profiles used in Figure 10. . . .". The discussion following this sentence seems to be related to Figure 10. There is no discussion on Figure 11. Fig. 11 caption says " . . .used in Figure 9". It should be Figure 10?

Page 12, line 401-405. "Figure 12 shows . . .approaches zero". There is not much discussion on Fig. 12. What does Figure 12 suggest? What is the definition of Delta theta_ent ? Which curve best represents observation? Does the figure mean that sigma_ext is sensitive or not sensitive to the entrained air properties?

Page 12, line 407-419. Does Table 3 include the entrainment sensitivity results from Figure 12?

---

## Referee Comment (RC3) · Anonymous Referee #3 · 25 May 2017

The manuscript presents an interesting study of aerosol-cloud-closure in terms of cloud CDNC and shortwave radiative flux using ground-based and UAV platform measurements, satellite retrievals at Mace Head, Ireland during summer 2015, as well as a 1-D aerosol-cloud parcel model simulations. The authors look at CDNC closure between Hoppel CDNC, satellite retrievals, and ACPM simulations, and cloud-top extinction and shortwave radiative flux closure between UAV measurements and ACPM simulations. The authors find that clouds in decoupled boundary layer have larger shortwave radiative flux differences between observations and simulations. More interestingly, the authors find that accounting for cloud-top entrainment in simulations greatly reduces the radiative flux differences. The manuscript is well written and organized. Overall,

the article is suitable for publication in the ACP with some revisions. Below are some specific comments.

Specific comments:

L77 and 86: the sentences are repeating.

Section "UAV vertical profiles": How cloud-top radiative fluxes are measured? It is not illustrated in the manuscript.

L205: need a reference here.

L260: Reference to Hoppel 1979 is not listed. I would suggest giving more details of using Dmin to estimate CDNC. How accurate is the estimation?

Figure 6: It is better to add variations of measured and satellite-retrieved CDNC. For comparisons between Dmin-estimated CDNC and simulated CDNC, they both use ground-based aerosol distribution measurements as input, therefore, these two are not independent.

L308: 0.3 or 0.5?

L326: Even for simulations with 50% decreased cloud-base aerosol, decoupled cases still have greater radiative differences than the coupled cases. Does that mean there are other factors other than aerosol between decoupled and coupled cases that contribute to the radiative differences?

---

## Referee Comment (RC4) · Anonymous Referee #4 · 1 Jun 2017

This paper presents results from a variety of measurements during an intensive field campaign at Mace Head in Ireland. It is perhaps unique in comparing estimates of cloud drop number concentration and radiative fluxes at cloud top based on several significantly different methods for a handful of cases during the campaign. Given the disparity among the cases (i.e. cumulus/stratocumulus; coupled/decoupled; adiabatic/sub-adiabatic), as well as the presentation of the results, it is a little unclear how to generalize the results of the study. The most substantive result seems to be the successful application of method for adjusting a parcel model calculation of the cloud-top radiative flux to account for dilution of the cloud by entrainment that results in a flux estimate that agrees better with in-situ measurements of cloud extinction. The paper

is appropriate for publication in ACP after addressing some minor revision.

In a couple of places some fairly arbitrary adjustments were made with inconclusive results. For example, in lines 319-322 the authors describe a test where the aerosol concentration imposed on the parcel model is arbitrarily reduced by 50% based on the notion that the aerosol concentration in the cloud layer of a decoupled boundary layer is likely to be less than what was measured at the surface. Yet the the change resulted in little change in the cloud-top radiative flux. How do the authors reconcile the small change in radiative flux for such a larger perturbation of the imposed aerosol concentration with their ultimate conclusion that the main source of error in their bottom-up radiative closure for the decoupled boundary layer cases is the lack of measurements to constrain the CCN concentration in the decoupled cloud layer?

In the conclusion it is stated that cloud-top entrainment is only observed on 2 out of 13 flight days, and a decoupled boundary layer on only 4 of 13 flight days. It might be valuable to include this in the abstract. While reading the paper, I was struggling to understanding the broader implications. Is there sufficient data to draw a tentative conclusion about the overall sign and/or magnitude of errors in bottom-up forcing calculations based on the surface station data at this location? If this can be addressed in any manner by the authors, then the paper will have substantially greater importance.

---

## Author Comment (AC1) · 13 Jul 2017

**Reviewer overview:**

The authors provide an analysis of cloud droplet closure using data collected at Mace Head, Ireland during summer 2015. The dataset includes surface based aerosol and remote sensing data from the Mace Head station. In addition, in situ vertical profile data was collected from a new UAV platform, which was deployed with a rotating payload comprising of meteorological probes, an aerosol optical sizing spectrometer and a cloud extinction monitor. Finally, the authors also make use of satellite cloud remote sensing products.

The authors conduct an aerosol-cloud microphysical closure analysis from the surface based data input into a parcel model (bottom-up) and from the satellite and in situ cloud extinction (top-down) to assess the uncertainty in deriving shortwave cloud radiative effects associated with microphysics. The authors find that when they account for reductions in cloud drop number concentration associated with entrainment, the difference between modelled and observed shortwave fluxes are reduced. The authors also find that decoupled clouds result in larger differences between modelled and observed shortwave fluxes, compared to well mixed cases.

Overall the paper is interesting and suitable for publication in ACP. I have a number of minor points listed below, which I urge the authors to consider before the paper is finalized.

**We thank the reviewer for their comments that significantly contributed to improving the original manuscript. Please see below responses to each of the authors comments and suggestions.**

**The authors want to note that values in Table 3 have slightly changed. These changes were brought about by reviewer #2's comment to present cloud optical thickness. It was noticed that the 'observed' optical thickness was not consistent between calculations that including and excluding cloud top entrainment. The observed optical thickness is calculated from the observed cloud droplet extinction. The observed droplet extinction is calculated by subtracting the simulated cloud droplet extinction and fitted difference in droplet extinction ($\delta\sigma_{ext}$) (Figure 8b,d,f). This was necessary to take into account the fact that the UAV's often missed portions of the cloud. The linear fit made it possible to fill the gaps. Since the observations should be consistent, the observations from the fit that excluded entrainment was compared to simulations with entrainment.**

General point: It might be useful to clarify in the abstract (and in sections before you define RF) that you are discussing shortwave radiative flux

**We have changed "radiative flux" to "shortwave radiative flux" in both the abstract and throughout the paper.**

L80 "surface latent heat flux, (i.e. evaporative cooling at the surface)" – this is misleading: surface latent heat flux does not induce cooling. It is independent of the heat budget at the surface. The mechanism, described in BW97, results in decoupling because under high LHF, there is a larger jump in buoyancy flux at cloud base, the cloud layer drives the turbulent motions and a zone of negative buoyancy flux develops in the sub-cloud layer. When this zone becomes too large it becomes dynamically favorable for the cloud layer to decouple from the sub-cloud layer.

**The text has been altered and is present with the response to the reviewer's comment on line 87.**

L81 BW97 claim that drizzle is not necessary for their "deepening-warming decoupling" mechanism, however they do show that it can have a substantial impact on the promotion of negative sub-cloud buoyancy fluxes and induce decoupling.

**The text has been altered and is present with the response to the reviewer's comment on line 87.**

L87 also related, moving air over a higher SST does not induce cooling. Suggest reviewing Stevens, 2002, Bretherton and Wyant, 1997 and Schubert et al., 1979 (not exhaustive list) for information about the mechanism of decoupling driven by increased surface latent heat fluxes and negative sub-cloud buoyancy fluxes.

**Based on the previous 3 comments and a review of the literature, this section has been re-written to more accurately describe the processes taking place that cause boundary layer decoupling:**

> **"Marine boundary layer decoupling is often seen in the tropics and has been attributed to processes that involve cloud heating from cloud-top entrainment, leading to decoupling of the boundary layer (Bretherton et al., 1997;Bates et al., 1998;Albrecht et al., 1995;Zhou et al., 2015;Stevens, 2002). In addition, Bretherton and Wyant (1997) have shown that the decoupling structure is mainly driven by a high latent heat flux that results in a large buoyancy jump across the cloud base. This high latent heat flux is attributed to easterlies bringing air over increasing SST, where the boundary layer becomes deeper and more likely to decouple (Albrecht et al., 1995). The cloud layer drives the turbulent motion and a zone of negative buoyancy flux develops below cloud. The turbulent motion is driven by radiative cooling at cloud top, causing air to sink (Lilly, 1968). The zone of negative buoyancy occurs because the deepening of the boundary layer causes the lifting condensation level of the updraft and downdraft to separate. This is important because latent heating in the cloud contributes significantly to the bouyancy in the cloud (Schubert et al., 1979). If this zone of negative buoyancy flux becomes deep enough, it is dynamically favorable for the cloud layer to become decoupled from the cloud layer (Bretherton et al., 1997). Bretherton and Wyant (1997) also show that drizzle can have a substantial impact on enhancing the negative buoyancy flux below cloud, but drizzle is not necessary for decoupling mechanism they proposed. Other factors, such as the vertical distribution of radiative cooling in the cloud, and sensible heat fluxes, play less important roles. Turton and Nicholls (1987) used a two-layer model to show that decoupling can also result from solar heating of the cloud layer; however, only during the day. Furthermore, Nicholls and Leighton (1986) showed observations of decoupled clouds with cloud-top radiative cooling and the resulting in-cloud eddies do not mix down to the surface (further suggesting radiative cooling plays a less important role). Russell et al. (1998) and Sollazzo et al. (2000) showed that, in a decoupled atmosphere the two distinct layers have similar characteristics (e.g., aerosol and trace gases composition), with different aerosol concentrations that gradually mix with each other, mixing air from the surface-mixed layer into the decoupled**

**layer and vice versa.  These previous studies also show that aerosol concentrations in the decoupled layer are lower than those in the surface-mixed layer implying an overestimation in cloud shortwave radiative flux when using ground-based aerosol measurements.**

L123 (sp) Nafion.

**The spelling has been corrected.**

L145-147 how is the scaling done? In Figure 7 and Figure 11, RH values are shown to be < 100% in the cloud layer.

**Referenced text: "As RH sensors are not accurate at high RH ( > 90%), the measured values have been scaled such that RH measurements are 100% in a cloud.  At altitudes where the UAV is known to be in-cloud (based on *in-situ* cloud extinction measurements) the air mass is considered saturated (RH ~ 100%)."**

**The calibration for RH values of 70-100% were adjusted (the slope of the calibration linear fit was modified) so that the maximum RH was 100%. The maximum RH before this correction which was typically between 90 and 95%. For the calculation of in-cloud water vapor content (for figure 10) RH values in cloud were recalibrated using cloud as 100% RH.  The simulation calculated RH values >100% in-cloud and therefore was not affected.**

L155 typo – Aerosols

**The typo is fixed.**

L204-206 Mixing state: can you clarify what you mean by "externally mixed types of particles". You then state that aerosols are internally mixed: is it fair to say that aerosols are internally mixed when this paper is discussing evidence of a significant fraction of air entrained into the boundary layer from above? Would aerosols from the free troposphere not have different chemical characteristics from the boundary layer? The phrase "lack of aerosol sources" is also ambiguous.

**The ACPM has the capability of including both internally and externally mixed particles. As indicated by the reviewer, the aerosols were internally mixed. We have removed "externally mixed types of particles" to avoid confusion.**

**Only parts of the cloud layer are suggested to have free tropospheric air entrained. Though the fraction of free tropospheric air in parts of the cloud are high, homogeneous entrainment would not result in the activation of new particles and therefore would not alter the cloud shortwave radiative forcing. Also, typically aerosols in the free troposphere are too small to be CCN active.**

**We have changed "owing to lack of aerosol sources" to "because there were no immediate strong sources of pollution".**

L215-226 Does the model include the effects of coalescence scavenging, which may be quite significant for a marine cloud over the 2-hour period given here.

**The model does not include the effects of coalescence scavenging. However, after looking further into our results, the simulation time is less than 20 minutes at the average updraft velocity, with the exception of the C21Cu case. Based on results from Feingold et al. [2013], coalescence scavenging rates are negligible for the CDNC and LWC (<0.4 g/m-3) for the case studies except for the C21Cu**

**case. The C21Cu case does have significantly high Liquid water content (>1.0 g/m-3), and therefore is susceptible to coalescence of droplets.**

**The following text has been added to clarify this point:**

**"==Feingold et al. (2013) showed that autoconversion and accretion rates are negligible for the modeled values of LWC and CDNC except for the C21Cu case, which had LWC > 1 g m-3. Thus, droplet number loss by collision coalescence can be neglected for all cases except for the C21Cu case.==" "**

**A footnote has been added to the table to indicate the C21Cu is susceptible to coalescence of droplets.**

L222 should there be a negative sign in your equation for the adiabatic cooling term (i.e. –gdz/cp)?

**Yes, we have added the negative sign to correct the equation.**

L340-342 I think you could be a bit clearer about how you come to this conclusion from the data shown in Table 2.

**Previous text: "For example, in the C11Sc case, in-situ observations do indeed show cloud-top inhomogeneous entrainment; consequently, the usual 30% reduction in CDNC does not need to be applied (Table 2)."**

**The text has been changed to the following to clearly indicate the reason for not applying the correction.**

**"==For the C11Sc case, before the correction, proposed by Freud et al. (2011), is applied the satellite derived CDNC (83 cm-3) is within 30% of the ACPM CDNC (88 cm-3) similar to the other cases (Figure 6), but if the correction is applied, the satellite derived CDNC (58 cm-3) is not within 30% of the ACPM CDNC. This indicates cloud top entrainment for the C11Sc case is== already ==inhomogeneous and== the usual 30% reduction in CDNC ==to correct for the inhomogeneous assumption== does not need to be applied. "**

L374-390 in both well-mixed and decoupled boundary layers, there are diabatic processes affecting the cloud layer namely, long-wave cooling of the cloud top, short wave absorption, drying due to drop sedimentation. To what extent do these processes interfere with the assumption of a cloud parcel being a mixture of cloud base air and entrained air?

**While these processes were not taken into account, they are expected to be small. The vertical extent of these clouds is small, consequently droplet diameters are relatively small (Reff < 15 microns) which limits the impact of droplet sedimentation. Typically, shortwave absorption is small and only slightly offsets long-wave cooling (Harrington et al. 1999). If long-wave cooling were the dominate process, the in-cloud lapse rate would be super-adiabatic. However, the in-cloud measured lapse rate was sub-adiabatic, so we conclude that entrainment warming was dominant mechanism in changing the in-cloud temperature. Also, long-wave cooling is greatest near the cloud top, meaning it is only important if a parcel remains near the cloud-top for a significant amount of time (Harrington et al. 1999; Hartman et al. 2004). For the entrainment cases considered in this study, the air masses have short residence times in the clouds (less than 20 minutes) and only spend a small fraction of this time at cloud-top.**

Fig 10: suggest putting the flight details in the caption (like Fig 11) for clarity

**We have added the case names to the description of figure 10 (like figure 11).**

L388-400 I think this section could be reworded to improve its clarity. I also have a few concerns: 1) It's not clear what you are referring to with the linear proportional relationship (L392). As you clarified, the qv=qt is only true outside the cloud, but if this mixing diagram is only used to illustrate processes in the cloud, what new information do you get for cloudy air with the addition of the second dimension (qv) over the 1D theta-E mixing calculation done with Eq.4? 2) The dashed line is linear by design, on a qt axis. Since qt=qv at the two end points these would indeed be the end points of the dashed line but on this qv axis the line would be curved 3) It is not clear what the adiabatic line is supposed to represent. Why does theta-E change during an adiabatic process?

**Original referenced text:**

> "Figure 10a and b present the relationships between two conservative variables measured by the UAV (water vapor content, $q_v$, and $\theta_e$) for C11Sc and D05Sc. The $q_v$ is derived from relative humidity measurements and is equivalent to the $q_t$ for sub-saturated, cloud-free air (i.e., < 100% RH).
>
> Figure 11 shows the relative humidity and $\theta_e$ profiles used in Figure 10. For both C11Sc and D05Sc, $\theta_{e,c}(z)$ is directly measured in-cloud, and $q_v$ and $\theta_e$ exhibit an approximately linear, proportional relationship (Figure 10; Eq. 4). The linear relationship is assumed to be a result of the cloud reaching a steady-state, with air coming from cloud-base and cloud-top (e.g. cloud lifetime >> mixing time). The observed in-cloud $q_v$ in Figure 10a and b is less than the conservative variable $q_t$, however, the figure also includes $q_t$ based on simulated adiabatic and cloud-top entrainment conditions. Eq. (4) is used to derive the simulated cloud-top entrainment conditions (Figure 10a and b), where the fraction entrained is used to calculate $q_t$ and shows a linear relationship between $q_t$ and $\theta_e$. Measurements above cloud-top (RH < 95%) with $q_v$ > 5.1 g kg$^{-1}$ and $q_v$ > 6.5 g kg$^{-1}$ are used to represent the properties of the entrained air for C11Sc and D05Sc, respectively (Figure 10)."

**Modified text:**

> "Figure 10a and b present the relationships between two conservative variables measured by the UAV (water vapor content, $q_v$, and $\theta_e$) for C11Sc and D05Sc. The $q_v$ is derived from relative humidity measurements and is equivalent to the $q_t$ for sub-saturated, cloud-free air (i.e., < 100% RH). The cloud-free air is shown in blue in Figure 10, where the below cloud measurements have lower $\theta_e$ than in-cloud and the above cloud measurements have higher $\theta_e$ than in-cloud.
>
> Figure 11 shows the relative humidity and $\theta_e$ profiles used in Figure 10. For both C11Sc and D05Sc, $\theta_{e,c}(z)$ is directly measured in-cloud, and $q_t$ and $\theta_e$ exhibit an approximately linear relationship (Figure 10; Eq. 4). The linear relationship of $q_t$ and $\theta_e$ (between the non-mixed sources of air indicated by orange circles in Figure 10) is assumed to be a result of the cloud reaching a steady-state, with air coming from cloud-base and cloud-top (e.g. cloud lifetime >> mixing time). The observed in-cloud $q_v$ in Figure 10a and b is less than the conservative variable $q_t$, however, the figure also includes $q_t$ based on simulated adiabatic (marked with an 'X') and cloud-top entrainment (dashed black line) conditions. Under adiabatic conditions $q_t$ and $\theta_e$ do not change in the cloud, which is why the adiabatic simulations only consists of one point in Figure 10. Eq. (4) is used to derive the simulated cloud-top entrainment conditions (Figure 10a and b), where the fraction entrained is used to calculate $q_t$ and shows

a linear relationship between $q_t$ and $\theta_e$. Measurements above cloud-top (RH < 95%), ==labeled entrained air==, with $q_v > 5.1$ g kg$^{-1}$ and $q_v > 6.5$ g kg$^{-1}$ are used to represent the properties of the entrained air for C11Sc and D05Sc, respectively (Figure 10). ==These conditions were chosen because these values are on the mixing line, between the non-mixed sources identified by the orange circles.=="

**Responses to each part of the comment:**

1. **The text now refers to the linear relationship in Figure 10: "The linear relationship of $q_t$ and $\theta_e$ (between the non-mixed sources of air indicated by orange circles in Figure 10) is assumed to be a result of the cloud reaching a steady-state, with air coming from cloud-base and cloud-top (e.g. cloud lifetime >> mixing time)."**
   **The qv is not necessary for equation 4, but the linear relationship between these 2 conservative variables in the cloud enables the visualization of a mixing line and enables us to show the change in total water content between adiabatic (without entrainment) and entrainment scenarios. Also, the linear relationship helps define which observations best represent entrained air (red points in figure 10).**

2. **The graph has now been modified so that the left axis represents observed qv and the right axis represents simulated qt.**

3. **$\theta_e$ should not change in an adiabatic process. Figure 10 has been modified so that the simulated $\theta_e$ in an adiabatic process does not change. The following text was added to the discussion of Figure 10: "Under adiabatic conditions $q_t$ and $\theta_e$ do not change in the cloud, which is why the adiabatic simulations only consists of one point in Figure 10."**

L417 what is the sensitivity of cloud extinction if mixing is homogeneous v.s. inhomogeneous compared to, say, the magnitude of the entrainment? Are there any other clues from your data set that could help confirm that the inhomogeneous process is a suitable assumption?

**We cannot calculate the degree to which entrainment was homogeneous with traditional methods because they involve cloud droplet size distributions observations, which were not possible with the class of UAVs used here. Nonetheless, previous observations (Burnet and Brenguier, 2007; Beals et al. 2015) have used cloud droplet size distribution observations to show cloud top entrainment is mostly inhomogeneous entrainment. The evaporation rate for homogeneous mixing strongly depends on mixing scales, so there is not a unique answer for homogeneous mixing (Lehmann et al. 2009).**

**Based on our results, the inhomogeneous correction used for the satellite measurements greatly increases the error in CDNC (when comparing to the ACPM CDNC) for the coupled entrainment case (C11Sc) suggesting the entrainment is inhomogeneous. Furthermore, inhomogeneous entrainment would result in greater CDNC and therefore, greater error in radiative flux.**

L470 What was happening on the other cases? Was the cloud layer more vigorously mixed, such that entrainment warming and drying was homogenized through the layer more rapidly?

**Referenced text: "…and decoupling of the boundary layer occurs on 4 of the 13 flight days."**

**The remaining 2 cases with a decoupled layer have insufficient in-cloud measurements for analysis and the clouds were too thin for satellite analysis. Figure 6 consist of the OPC concentration profile from one of these 2 cases and has a cloud thickness of less than 50 m.**

**A parenthetical statement was added to the referenced text:**

**"and decoupling of the boundary layer occurs on four of the 13 flight days (two decoupled cloud cases were not discussed due to the lack of in-cloud measurements)."**

L474 "presence (of) marine biogenic. . ."

**We have added the word "of".**

L474 local anthropogenic. . .what?

**The sentence was removed since the focus of the paper is marine boundary layer observations and not anthropogenic sources.**

L475 "observations and simulat(ed)"?

**We have changed the word "modeled" to "simulated" as suggested by the reviewer.**

---

## Author Comment (AC2) · 13 Jul 2017

**Reviewer overview:**

Summary: This manuscript presents an observational analysis to understand sources of uncertainty in deriving cloud radiative flux. The observations are from a number of platforms, including ground based, UAV, and satellite measurements. They used a 1-D microphysical model in conjunction with observations to derive microphysical and optical properties of observed clouds. The differences were found in radiative fluxes between the simulated and the observed. They concluded that the cloud-top entrainment is an important source of uncertainty for the cloud radiative flux calculation; it is particularly true for decoupled cloud boundary layers because ground-based measurements are no longer enough to obtain reliable data in the decoupled cloud layer. Authors' overall analysis technique is good and their conclusion is important and interesting. My main criticism is that some discussions and figures are not clear and confusing. I recommend publication after following comments are addressed.

**We thank the reviewer for their comments that significantly contributed to improving the original manuscript. Please see below responses to each of the authors comments and suggestions.**

**The authors want to note that values in Table 3 have slightly changed. These changes were brought about by reviewer #2's comment to present cloud optical thickness. It was noticed that the 'observed' optical thickness was not consistent between calculations that including and excluding cloud top entrainment. The observed optical thickness is calculated from the observed cloud droplet extinction. The observed droplet extinction is calculated by subtracting the simulated cloud droplet extinction and fitted difference in droplet extinction ($\delta\sigma_{ext}$) (Figure 8b,d,f). This was necessary to take into account the fact that the UAV's often missed portions of the cloud. The linear fit made it possible to fill the gaps. Since the observations should be consistent, the observations from the fit that excluded entrainment was compared to simulations with entrainment.**

I am wondering about the significance of showing the cloud-top extinction in Table 2 and 3. Even though the cloud-top radiative flux differences (Delta FR) in the two decoupled cases are larger than those in the coupled cases, delta sigma_ext values are similar for all the cases as shown in Table 3. The cloud-top value delta sigma_ext doesn't seem to mean a lot in terms of cloud optical property. Because the cloud-top radiative flux (RF) depends on the optical depth as shown in (2), it is probably more appropriate to show cloud optical depth (tau).

**Cloud optical depth has been added to Tables 2 and 3.**

Page 2, line 71: "Such decoupled layers often contain two distinct cloud layers, . . . a lower layer within the well-mixed surface layer and a higher decoupled residual layer between the free atmosphere and surface layer". I don't think the surface layer can be well mixed because turbulent eddies there are too small near the surface to produce strong mixing. You probably meant surface based mixed layer. That is,

a mixed layer that is connected to, but deeper than the surface layer. Why do you call a decoupled layer "residual layer"? Is there turbulence source in the decoupled layer? Does it have clouds?

**We have modified the text to say "surface mixed layer". We have also changed "residual layer" to "decoupled layer". The decoupled layer can have clouds and therefore a source of turbulence which is described by the following text that has been added:**

==**"The cloud layer drives the turbulent motion and a zone of negative buoyancy flux develops below cloud. The turbulent motion is driven by radiative cooling at cloud top, causing air to sink [*Lilly et al.*, 1968]."**==

Page 3, line 75: "the surface mixed layer". Surface based mixed layer?

**We have chosen to use "surface mixed layer" to define the lower layer in a decoupled boundary layer. This is consistent with a previous Mace Head paper on decoupling boundary layers (Milroy et al. 2011)**

Page 3, line 77 and line 80: " . . . involve cloud heating and surface cooling" and " i.e., evaporative cooling at the surface" I am not sure what is meat by the "surface cooling" or "evaporative cooling". Note that the surface evaporative cooling by surface moisture flux only cools the ocean surface, not the sub-cloud layer. I do not think the "surface evaporative cooling" directly contributes to the decoupling. Could you give a bit more explanation on this? An increase in the moisture flux with increasing SST enhances the cloud layer buoyancy flux, which intensifies the cloud-top entrainment to mix warmer and drier air into clouds, leading to negative buoyancy flux below cloud base.

**The text in this section has been largely modified to more accurately explain the processes. The text has been restated in the response to reviewer 1 and is also shown below:**

> **"Marine boundary layer decoupling is often seen in the tropics and has been attributed to processes that involve cloud heating from cloud-top entrainment, leading to decoupling of the boundary layer (Bretherton et al., 1997;Bates et al., 1998;Albrecht et al., 1995;Zhou et al., 2015;Stevens, 2002). In addition, Bretherton and Wyant (1997) have ==shown== that the decoupling structure is mainly driven by a ==high latent heat flux that results in a large buoyancy jump across the cloud base. This high latent heat flux is attributed to easterlies bringing air over increasing SST, where the boundary layer becomes deeper and more likely to decouple (Albrecht et al., 1995). The cloud layer drives the turbulent motion and a zone of negative buoyancy flux develops below cloud. The turbulent motion is driven by radiative cooling at cloud top, causing air to sink (Lilly, 1968). The zone of negative buoyancy occurs because the deepening of the boundary layer causes the lifting condensation level of the updraft and downdraft to separate. This is important because latent heating in the cloud contributes significantly to the bouyancy in the cloud (Schubert et al., 1979). If this zone of negative buoyancy flux becomes deep enough, it is dynamically favorable for the cloud layer to become decoupled from the cloud layer (Bretherton et al., 1997). Bretherton and Wyant (1997) also show that drizzle can have a substantial impact on enhancing the negative buoyancy flux below cloud, but drizzle is not necessary for decoupling mechanism they proposed.== Other factors, such as the vertical distribution of radiative cooling in the cloud, and sensible heat fluxes, play less important roles. Turton and Nicholls (1987) used a two-layer model to show that decoupling can also result from solar heating of the cloud layer; ==however, only during the day. Furthermore, Nicholls and Leighton (1986) showed observations of decoupled clouds with== cloud-top radiative cooling and the resulting ==in-cloud== eddies do not mix down to the surface ==(further suggesting radiative cooling plays a less important role).== Russell et al. (1998)  and Sollazzo et al. (2000)**

**showed that, in a decoupled atmosphere the two distinct layers have similar characteristics (e.g., aerosol and trace gases composition), with different aerosol concentrations that gradually mix with each other, mixing air from the surface-mixed layer into the decoupled layer and vice versa. These previous studies also show that aerosol concentrations in the decoupled layer are lower than those in the surface-mixed layer implying an overestimation in cloud shortwave radiative flux when using ground-based aerosol measurements. "**

Page 8, line 281-282 about Figure 8. Could you put the flight code (D05Sc, C11Sc and C21Cu) inside the plot boxes? That would be easy to see. The caption of Figure 8 mentions the difference between UAV-observed (green measurements) and ACPMsimulated (black line) to calculate delta sigma_ext. But it looks like you also calculate the cloud free values too. Although the (a)-(f) are labeled in each plot, they are not used in the caption.

**The flight code has been put inside the plot boxes. We have removed "(green measurements)" since we do calculate delta sigma_ext for cloud free values as the reviewer has pointed out. We have also included the letters in the caption to refer to each plot in the figure.**

Page 10, line 354-357: "The UAV observations show both C11Sc have sub-adiabatic lapse rate measurements, compared to simulated moist-adiabatic lapse rates within the cloud (Table 2). . ... The sub-adiabatic lapse rate is attributed to cloud-top entrainment . . ... at cloud-top (e.g., Figure 7a)" Where is the comparison between the observed and simulated lapse rate? I only see the simulated values in Table 2. Could you draw a line in Figure 7a to show the adiabatic lapse rate? It is hard to see the lapse rate is sub-adiabatic

**The sub-adiabtic lapse-rate results are now expressed in the text rather than the table because there were only sub-adiabatic lapse rates for two of the cases. Table 2 is cited to show the measured and simulated lapse rate.**

**The following text, at the end of section 3.2, compares $\delta$RF when using the adiabatic lapse rate and the observed lapse rate (now refered to as the lapse rate adjustment entrainment method):**

**"Finally, the lapse rate adjustment entrainment method [Sanchez et al., 2016] does improve ACPM accuracy between in-situ and satellite-retrieved cloud optical properties relative to the adiabatic simulations, but has greater $\delta\sigma_{ext}$ throughout the cloud than the inhomogeneous mixing entrainment method. For the lapse rate adjustment entrainment method $\delta$RF decreased from 88 Wm$^{-2}$ to 61 Wm$^{-2}$ and 48 Wm$^{-2}$ to 32 Wm$^{-2}$ for D05Sc and D11Sc respectively."**

**We have not added a line to show the adiabatic lapse rate to in figure 7a because the line, with a 1 K km$^{-1}$ greater lapse rate, would not be noticeably different than the measured lapse rate due to the large x-axis range. The reference to Figure 7a has been removed.**

Page 11-12, 391-399: "For both C11Sc and D05SC,. . .. . . exhibit an approximately linear, proportional relationship (Figure10; Eq. 4.) . . . ". This paragraph is a bit confusing. What flights do those curves come from in Fig. 10? Could you state clearly which part you were referring to that is linear? In Fig. 10, the cloudy part (green curve) is not linear because qv is not conserved variable for condensation/evaporation process.

**The following text has been modified to indicate $q_t$ and $\theta_e$ have a linear relationship, and that it is shown between the two orange circles:**

**"For both C11Sc and D05Sc, $\theta_{e,c}(z)$ is directly measured in-cloud, and $q_t$ and $\theta_e$ exhibit an approximately linear relationship (Figure 10; Eq. 4). The linear relationship of $q_t$ and $\theta_e$ (between the non-mixed sources of air indicated by orange circles in Figure 10) is assumed to be a result of the cloud reaching a steady-state, with air coming from cloud-base and cloud-top (e.g. cloud lifetime >> mixing time)."**

**The flight codes are added to figure 10.**

What is meant by "entrained air"? Does it consist of both free air and turbulent air or only free atmosphere and non-mixed air? Does it contain any cloud droplets? If not, why is it (red curve) not linear, particularly for the top panel plot?

**The entrained air is the air that is mixed into cloud top which is the air directly above the cloud (within 100 m) and do not contain cloud droplets. The air directly above the cloud may or may not be the free troposphere. For example, in the bottom panel of figure 10, the points in between the 2 circles represent the mixed air layer that you have referred to. Though this air is not necessarily from the free troposphere, it is what will mix with the cloud top. A point in the orange circle (Figure 10) could have been used to represent pure free tropospheric air that would entrain into the cloud, however using the red points in the mixed air yields the same result because it is on the mixing line and they are more physical representation to use since these are directly above the cloud. The entrained fraction (X in equation 5) will change, but approximately the same amount of liquid water will evaporate no matter which point is used on this mixing line for the entrained air properties. We have changed "entrained air sources" to "entrained air properties used in simulations" in the figure caption.**

**The red curve appears not to be linear (in the top panel of figure 10) mainly because the mixed air (between the two orange circles in Figure 10) has a smaller layer with no cloud so essentially the line is shorter. It is also possible that the UAV partially re-entered the very top of the cloud momentarily, causing an increase in RH even though $\sigma_{ext}$ does not increase because the change is below the detection limit. Also, as mentioned in the manuscript the RH sensor is not particularly accurate when RH is greater than 90%, and the water vapor content (y axis of figure 10) is calculated from the RH. The variability in the entrained water vapor is included in the errors in Table 3.**

What is the flight code (or number) for these two plots in Fig. 10? Please identify the blue dashed line in the text when discussing the entrainment conditions. There is no (a) and (b) in Fig 10. "Measurements above cloud-top (RH < 95%) with qv > 5.1 g kg-1 and qv > 6.5 g kg-1 are used to represent the properties of the entrained air". How do you choose this criterion for the entrained air? You should specify clearly the properties of the non-mixed sources of air: what are the values of theta_e and q_v of the air source? The orange circles include too many possibilities of these values.

**The flight code has been added to the figure.**

**We have now indicated the simulated adiabatic and entrainment conditions in the text:**

**"The observed in-cloud qv in Figure 10a and b is less than the conservative variable qt, however, the figure also includes qt based on simulated adiabatic (marked with an 'X') and cloud-top entrainment (dashed black line) conditions."**

**Blue cloud-free air (blue points) are now mentioned with the addition of the following sentence:**

**"The cloud-free air is shown in blue in Figure 10, where the below cloud measurements have lower θe than in-cloud and the above cloud measurements have higher θe than in-cloud."**

**(a) and (b) have been added to figure 10.**

**The quoted text has been supplemented to include the criteria for choosing entrained air: "Measurements above cloud-top (RH < 95%), labeled entrained air, with $q_v$ > 5.1 g kg$^{-1}$ and $q_v$ > 6.5 g kg$^{-1}$ are used to represent the properties of the entrained air for C11Sc and D05Sc, respectively (Figure 10). These conditions were chosen because these values are on the mixing line, between the non-mixed sources identified by the orange circles."**

**The properties of the entrained air (theta_e and q_v) are given by the red "entrained air" points in Figure 10. The orange circles are not meant to define values, but simple point out approximate end points to the mixing line. As stated in the response to the previous comment, using the properties of the "entrained air", shown in red, is equivalent to using the an observation from the top of this mixed layer.**

Line 391: "Figure 11 shows the relative humidity and theta_e profiles used in Figure 10. . . .". The discussion following this sentence seems to be related to Figure 10. There is no discussion on Figure 11. Fig. 11 caption says " . . .used in Figure 9". It should be Figure 10?

**The main point of figure 11 was to show the measurements used to make figure 10 as a vertical profile.**

**The figure 11 caption reference to figure 9 has been changed to figure 10.**

Page 12, line 401-405. "Figure 12 shows . . .approaches zero". There is not much discussion on Fig. 12. What does Figure 12 suggest? What is the definition of Delta theta_ent ? Which curve best represents observation? Does the figure mean that sigma_ext is sensitive or not sensitive to the entrained air properties?

**The figure caption has been changed to the following to define delta theta_ent and delta q_t:**

**"Figure 12. Sensitivity of simulated cloud extinction based on variability of entrained air potential temperature ($\theta$ent, K) and entrained air total water mixing ratio (qt,ent, g kg-1) for the C11Sc case. The $\Delta\theta$ent and $\Delta$qt,ent terms define the change in the entrained $\theta$ and qt values where no change ($\Delta\theta$ent = 0 and $\Delta$qt,ent = 0) is equivalent to the adiabatic simulation with entrainment from Figure 8c."**

**The intent with Figure 12 was not to fit the data, but instead show how the sensitive the simulated droplet extinction is to changes in properties of the entrained air. The sigma_ext is not very sensitive to the entrainment properties that were measured, but under different circumstances (lower $\theta$ and q$_t$) sigma_ext can be very sensitive.**

**The last sentence has been added to the quoted text to clarify the connection with Figure 12 and equation 5:**

**"Figure 12 shows the sensitivity of the simulated cloud extinction profile, for the 11 August case, based on measurement uncertainties related to the entrained qt and $\theta$. The key variable for identifying the entrained fraction (Eq. 5), $\theta_{e,ent}$, is a function of qt and $\theta$, so a decrease in either parameter results in a proportional decrease in $\theta_{e,ent}$. Eq. (5) shows that entrainment fraction becomes more sensitive to the uncertainty related to the measurement of $\theta_e$ as the difference between $\theta_{e,ent}$ and $\theta_{e,CB}$ approaches zero. This is also shown in Figure 12 where sigma_ext is more sensitive to lower entrained qt and $\theta$ values."**

Page 12, line 407-419. Does Table 3 include the entrainment sensitivity results from Figure 12?

**Yes, the errors given in Table 3 account for the range of $\theta_{e,ent}$ and $q_{t,ent}$ measured (red points in figure 10).**

---

## Author Comment (AC3) · 13 Jul 2017

The manuscript presents an interesting study of aerosol-cloud-closure in terms of cloud CDNC and shortwave radiative flux using ground-based and UAV platform measurements, satellite retrievals at Mace Head, Ireland during summer 2015, as well as a 1-D aerosol-cloud parcel model simulations. The authors look at CDNC closure between Hoppel CDNC, satellite retrievals, and ACPM simulations, and cloud-top extinction and shortwave radiative flux closure between UAV measurements and ACPM simulations. The authors find that clouds in decoupled boundary layer have larger shortwave radiative flux differences between observations and simulations. More interestingly, the authors find that accounting for cloud-top entrainment in simulations greatly reduces the radiative flux differences. The manuscript is well written and organized. Overall, the article is suitable for publication in the ACP with some revisions. Below are some specific comments.

**We thank the reviewer for their comments. Please see below responses to each of the authors comments and suggestions.**

**The authors want to note that values in Table 3 have slightly changed. These changes were brought about by reviewer #2's comment to present cloud optical thickness. It was noticed that the 'observed' optical thickness was not consistent between calculations that including and excluding cloud top entrainment. The observed optical thickness is calculated from the observed cloud droplet extinction. The observed droplet extinction is calculated by subtracting the simulated cloud droplet extinction and fitted difference in droplet extinction ($\delta\sigma_{ext}$) (Figure 8b,d,f). This was necessary to take into account the fact that the UAV's often missed portions of the cloud. The linear fit made it possible to fill the gaps. Since the observations should be consistent, the observations from the fit that excluded entrainment was compared to simulations with entrainment.**

Specific comments:

L77 and 86: the sentences are repeating.

**Referenced text:**

> **"Marine boundary layer decoupling is often seen in the tropics and has been attributed to processes that involve cloud heating and surface cooling as cloud warming can result from cloud-top entrainment, leading to decoupling of the boundary layer [*Albrecht et al.*, 1995; *Bates et al.*, 1998; *Bretherton et al.*, 1997]. In addition, Bretherton and Wyant [1997] have suggested that the decoupling structure is mainly driven by an increasing ratio of the surface latent heat flux, (i.e., evaporative cooling at the surface) to the net radiative cooling within the cloud, while other factors, such as drizzle, the vertical distribution of radiative cooling in the cloud, and sensible heat fluxes, play less important roles. Turton and Nicholls [1987] used a two-layer model to show that decoupling can also result from solar heating of the cloud layer. Nicholls and Leighton [1986] suggested decoupling results from cloud-top**

radiative cooling and the resulting eddies do not mix down to the surface. Zhou et al. [2015] showed that the entrainment of the dry warm air above the inversion could also be the cause. Marine boundary layer decoupling is often seen in the tropics and has been attributed to easterlies bringing air over increasing SST, which increases latent cooling and adds negative buoyancy below the cloud layer [*Albrecht et al.*, 1995]."

The text has been modified to the following based on responses from reviewers 1-3 and previously restated in responses to reviewer 1 and 2:

"Marine boundary layer decoupling is often seen in the tropics and has been attributed to processes that involve cloud heating from cloud-top entrainment, leading to decoupling of the boundary layer (Bretherton et al., 1997;Bates et al., 1998;Albrecht et al., 1995;Zhou et al., 2015;Stevens, 2002). In addition, Bretherton and Wyant (1997) have shown that the decoupling structure is mainly driven by a high latent heat flux that results in a large buoyancy jump across the cloud base. This high latent heat flux is attributed to easterlies bringing air over increasing SST, where the boundary layer becomes deeper and more likely to decouple (Albrecht et al., 1995). The cloud layer drives the turbulent motion and a zone of negative buoyancy flux develops below cloud. The turbulent motion is driven by radiative cooling at cloud top, causing air to sink (Lilly, 1968). The zone of negative buoyancy occurs because the deepening of the boundary layer causes the lifting condensation level of the updraft and downdraft to separate. This is important because latent heating in the cloud contributes significantly to the bouyancy in the cloud (Schubert et al., 1979). If this zone of negative buoyancy flux becomes deep enough, it is dynamically favorable for the cloud layer to become decoupled from the cloud layer (Bretherton et al., 1997). Bretherton and Wyant (1997) also show that drizzle can have a substantial impact on enhancing the negative buoyancy flux below cloud, but drizzle is not necessary for decoupling mechanism they proposed. Other factors, such as the vertical distribution of radiative cooling in the cloud, and sensible heat fluxes, play less important roles. Turton and Nicholls (1987) used a two-layer model to show that decoupling can also result from solar heating of the cloud layer; however, only during the day. Furthermore, Nicholls and Leighton (1986) showed observations of decoupled clouds with cloud-top radiative cooling and the resulting in-cloud eddies do not mix down to the surface (further suggesting radiative cooling plays a less important role)."

Section "UAV vertical profiles": How cloud-top radiative fluxes are measured? It is not illustrated in the manuscript.

There were no airborne direct measurements of cloud-top radiative flux. Cloud-top radiative flux is calculated using extinction measurements from the cloud droplet sensor measurements and from ACPM simulations. The cloud albedo is calculated from extinction (equations 1-3) and the albedo is used to calculate the cloud-top radiative flux. The following text in section 2.4 explains how the cloud-top shortwave radiative flux is calculated: "the shortwave radiative flux (RF) is calculated as $RF = \alpha Q$, where Q is the daily-average insolation at Mace Head and $\alpha$ is the cloud albedo."

In the "UAV vertical profiles" section the last sentence of the following text was added for clarity: "In-cloud extinction was measured in-situ using a miniature optical cloud droplet sensor developed at the University of Reading  [Harrison and Nicoll, 2014]. The sensor operates by a backscatter principle using modulated LED light which is backscattered into a central photodiode.  Comparison

of the sensor with a Cloud Droplet Probe (DMT) demonstrate good agreement for cloud droplet diameters >5μm [Nicoll et al., 2016]. ==The extinction measurements were used to calculate cloud-top shortwave radiative flux and is further discussed in section 2.4.==”

L205: need a reference here.

**A reference is included at the end of the sentence: “The model employs a dual moment (number and mass) algorithm to calculate particle growth from one size section to the next for non-evaporating compounds (namely, all components other than water) using an accommodation coefficient of 1.0 [Raatikainen et al., 2013].”**

L260: Reference to Hoppel 1979 is not listed. I would suggest giving more details of using Dmin to estimate CDNC. How accurate is the estimation?

**The Hoppel reference has been added.**

**The last sentence in the following text has been added to explicitly explain how to calculate the Hoppel CDNC: “The dry aerosol particles with diameters greater than the Hoppel Dmin have undergone cloud processing and are used here to estimate the CDNC. For each of the case study days, Figure 5 demonstrates the aerosol size distribution measurements, from the SMPS and APS, that are used to find the Hoppel Dmin, Hoppel CDNC and used to initialize the ACPM. ==The Hoppel CDNC is calculated by integrating the SMPS and APS combined size distributions for aerosol sizes greater than Hoppel Dmin.==”**

**The Hoppel CDNC is within 30% of both the simulated CDNC and the satellite estimated CDNC.**

Figure 6: It is better to add variations of measured and satellite-retrieved CDNC. For comparisons between Dmin-estimated CDNC and simulated CDNC, they both use ground-based aerosol distribution measurements as input, therefore, these two are not independent.

**We do not have measured CDNC, but instead are using the CDNC calculated by the aerosol-cloud parcel model (ACPM). Even though the Dmin-estimated CDNC and simulated CDNC both use ground-based measurements of the aerosol distributions, the ACPM simulates the supersaturation to determine the critical diameter based on the size and chemical composition of the particles. The critical diameter is not necessarily the same as the Dmin diameter. The ACPM is the main link between observations and the satellite measurement, which is why both the satellite CDNC and Dmin-estimated CDNC are compared to the ACPM CDNC. The main purpose of the figure was to show that the satellite CDNC are within 30% of the ACPM CDNC because the error associated with the satellite retrieval method is 30% (Rosenfeld et al., 2016).**

L308: 0.3 or 0.5?

**The minimum diameter of the OPC is 0.3 microns. This has been corrected in the manuscript.**

L326: Even for simulations with 50% decreased cloud-base aerosol, decoupled cases still have greater radiative differences than the coupled cases. Does that mean there are other factors other than aerosol between decoupled and coupled cases that contribute to the radiative differences?

**The main reason the radiative flux difference is large is simply because the cloud (D05Sc) is the thinnest cloud, and therefore error’s in extinction (from measurement error or error in simulated) have a larger influence on the radiative differences. From equation 2, a small change in a cloud with low optical thickness (thin cloud) has a greater effect on the albedo than a small change in a**

**high optical thickness (thick cloud). Notice the error in extinction for the D05Sc case in table 2 is similar to the C11Sc case even though the error in RF is lower for C11Sc.**

---

## Author Comment (AC4) · 13 Jul 2017

This paper presents results from a variety of measurements during an intensive field campaign at Mace Head in Ireland. It is perhaps unique in comparing estimates of cloud drop number concentration and radiative fluxes at cloud top based on several significantly different methods for a handful of cases during the campaign. Given the disparity among the cases (i.e. cumulus/stratocumulus; coupled/decoupled; adiabatic/sub-adiabatic), as well as the presentation of the results, it is a little unclear how to generalize the results of the study. The most substantive result seems to be the successful application of method for adjusting a parcel model calculation of the cloud top radiative flux to account for dilution of the cloud by entrainment that results in a flux estimate that agrees better with in-situ measurements of cloud extinction. The paper is appropriate for publication in ACP after addressing some minor revision.

**We thank the reviewer for their comments. Please see below responses to each of the authors comments and suggestions.**

**The authors want to note that values in Table 3 have slightly changed. These changes were brought about by reviewer #2's comment to present cloud optical thickness. It was noticed that the 'observed' optical thickness was not consistent between calculations that including and excluding cloud top entrainment. The observed optical thickness is calculated from the observed cloud droplet extinction. The observed droplet extinction is calculated by subtracting the simulated cloud droplet extinction and fitted difference in droplet extinction ($\delta\sigma_{ext}$) (Figure 8b,d,f). This was necessary to take into account the fact that the UAV's often missed portions of the cloud. The linear fit made it possible to fill the gaps. Since the observations should be consistent, the observations from the fit that excluded entrainment was compared to simulations with entrainment.**

In a couple of places some fairly arbitrary adjustments were made with inconclusive results. For example, in lines 319-322 the authors describe a test where the aerosol concentration imposed on the parcel model is arbitrarily reduced by 50% based on the notion that the aerosol concentration in the cloud layer of a decoupled boundary layer is likely to be less than what was measured at the surface. Yet the the change resulted in little change in the cloud-top radiative flux. How do the authors reconcile the small change in radiative flux for such a larger perturbation of the imposed aerosol concentration with their ultimate conclusion that the main source of error in their bottom-up radiative closure for the decoupled boundary layer cases is the lack of measurements to constrain the CCN concentration in the decoupled cloud layer?

**Figure 9 shows the OPC concentration reduced by almost 50% in decoupled layer (compared to the surface based mixed layer), though this is not the same case. The choice of 50% was loosely based on this given there were no other measurements to base this choice on. We have now referred to Figure 9 in the text:**

**"ACPM simulations were conducted using aerosol concentrations based on the approximate average decoupled to coupled aerosol concentration ratio (50%, Figure 9) to estimate the difference in shortwave radiative flux. "**

Previous literature has shown there are cases were CDNC is sensitive to aerosol concentration (aerosol limited) while others are sensitive to updraft velocity (updraft limited). The manuscript discusses the results of decreasing the aerosol concentrations in simulations of both the D05Sc and D06Cu cases. The D06Cu case which has a large range of updraft velocities (0-1.6 m/s) had significantly fewer (42%) CDNC after reducing the aerosol concentration. The D05Sc has significantly lower updraft velocities, ranging from 0-0.3 m s-1, and therefore, is updraft limited. The CDNC is very sensitive at these low updraft velocities, so it is likely that the combined modeled updraft resolution of 0.1 m s-1 and error in updraft velocity measurements is the cause for the large error in shortwave radiative forcing ($\delta$RF) of 33 W m-2 (Table 2) for the D05Sc case, after accounting for cloud top entrainment.

The following text has been changed to incorporate this information:

"For the D05Sc case, simulations with 50% decreased cloud-base aerosol concentrations show only slight differences in $\delta$RF of 2 Wm$^{-2}$ and decreases in CDNC of 10%. The decrease in aerosol concentration resulted in increased supersaturation due to the low water uptake from fewer activating droplets. The increased supersaturation caused smaller aerosols to activate (Raatikainen et al., 2013) and therefore, little change in CDNC. The D05Sc case has very low updraft velocities (0-0.3 m s$^{-1}$). At low updraft velocities, the CDNC is often updraft limited (Reutters et al., 2009). This means the CDNC is very sensitive to the updraft velocities and less sensitive to aerosol concentration. Small errors in updraft velocity and low modeled updraft resolution (0.1 m s$^{-1}$) likely contributes significantly to the error in this case. The D06Cu was not influenced as much by low water uptake because the CDNC was much higher at 171 cm$^{-3}$ compared to 86 cm$^{-3}$ for D05Sc. The D06Cu the CDNC decreased by 42% and $\delta$RF decreased by 18 Wm$^{-2}$. The updraft velocity range for the D06Cu case is significantly higher than the D05Cu case (0-1.6 m s$^{-1}$). The higher velocities for the D05Sc and greater sensitivity to aerosol concentration suggest this case is aerosol limited (Reutters et al., 2009). Both decoupled cases still have a $\delta$RF greater than the coupled cases."

For the D06Cu case, the 42% decrease in CDNC, significantly reduced $\delta$RF from 74 to 56 w m-2. A $\delta$RF of 56 w m-2 is still high compared to the decoupled cases. It is possible that the difference in aerosol concentration between the coupled and decoupled boundary layer is greater than 50%. We do not have aerosol concentration measurements in the decoupled layer for this case. Also, it is possible that this case experienced some cloud top entrainment. The measured lapse rate for this case was slightly higher (0.1 K km-1) than the adiabatic lapse rate, however this was within instrument error, so cloud top entrainment was not explored. If the heating is offset by long wave cooling (not considered in this paper), then the effect of entrainment may be significant. Note, the two entrainment cases studied both had measured lapse rates that were 1 K km-1 higher than the adiabatic lapse rate.

The following text has been changed to incorporate this information:

"The UAV observations show that both C11Sc and D05Sc have sub-adiabatic lapse rate measurements, compared to simulated moist-adiabatic lapse rates within the cloud (Table 2). The difference between the observed and simulated lapse rates therefore suggests a source of heating in the cloud. The sub-adiabatic lapse rate is attributed to cloud-top entrainment by downward mixing of warmer air at cloud-top. The D06Cu case has a slightly sub-adiabatic observed lapse rate (Table 2), however the difference with respect to an adiabatic lapse rate is within instrument error. For this reason, cloud top entrainment is not explored for this case, though it may contribute slightly to the error."

In the conclusion it is stated that cloud-top entrainment is only observed on 2 out of 13 flight days, and a decoupled boundary layer on only 4 of 13 flight days. It might be valuable to include this in the abstract. While reading the paper, I was struggling to understanding the broader implications. Is there sufficient data to draw a tentative conclusion about the overall sign and/or magnitude of errors in bottom-up forcing calculations based on the surface station data at this location? If this can be addressed in any manner by the authors, then the paper will have substantially greater importance.

**After revisiting the statement (that cloud-top entrainment is only observed on 2 out of 13 flight days, and a decoupled boundary layer on only 4 of 13 flight days) we have decided to reworded this statement to more clearly what these statistics are based on:**

**"Based on airborne observations with UAVs, decoupling of the boundary layer occurred on four of the 13 flight days (two decoupled cloud cases were not discussed due to the lack of in-cloud measurements). However, cloud drop entrainment was only observed on two of those days, limited by the ability to make in-situ measurements. These measurements occurred during the summer, so additional measurements are needed to look at seasonal trends."**

**Because the entrainment statistic is limited by measurement capabilities we have decided not to include this in the abstract.**

**The main broader implications of these results are that cloud-top entrainment and decoupling of the boundary layer lead to over estimation of cloud-top shortwave radiative forcing when using the adiabatic and well mixed boundary layer assumptions, respectively. While we have indicated the magnitude of these errors for the cases presented, there are only a limited number of cases in this manuscript to draw statistics on the occurrence of these scenarios. In order have a Many more case studies are needed to conclude more specific implications for the Mace Head location. Furthermore, similar studies at other locations are necessary to understand global implications.**